# LoRAShop: Training-Free Multi-Concept Image Generation and Editing with Rectified Flow Transformers

**Yusuf Dalva**     **Hidir Yesiltepe**     **Pinar Yanardag**
{ydalva, hidir, pinary}@vt.edu
Virginia Tech
lorashop.github.io

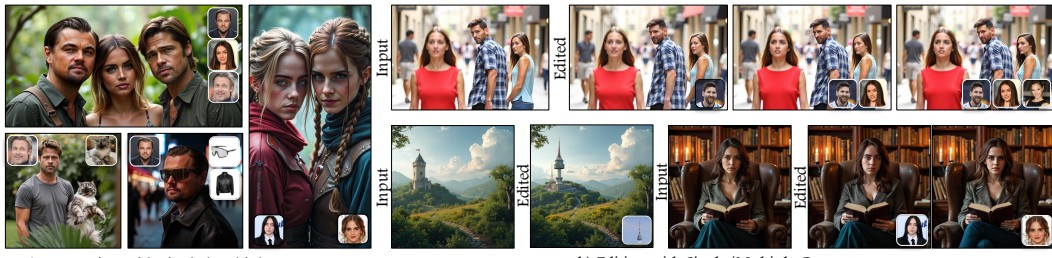

a) Generation with Single/Multiple Concepts        b) Editing with Single/Multiple Concepts

Figure 1: **LoRAShop.** We present LoRAShop, a training-free framework enabling the simultaneous use of multiple LoRA adapters for generation and editing. By identifying the coarse boundaries of personalized concepts as subject priors, we allow the use of multiple LoRA adapters by eliminating the "cross-talk" between different adapters.

## Abstract

We introduce LoRAShop, the first framework for multi-concept image editing with LoRA models. LoRAShop builds on a key observation about the feature interaction patterns inside Flux-style diffusion transformers: concept-specific transformer features activate spatially coherent regions early in the denoising process. We harness this observation to derive a disentangled latent mask for each concept in a prior forward pass and blend the corresponding LoRA weights only within regions bounding the concepts to be personalized. The resulting edits seamlessly integrate multiple subjects or styles into the original scene while preserving global context, lighting, and fine details. Our experiments demonstrate that LoRAShop delivers better identity preservation compared to baselines. By eliminating retraining and external constraints, LoRAShop turns personalized diffusion models into a practical 'photoshop-with-LoRAs' tool and opens new avenues for compositional visual storytelling and rapid creative iteration.

## 1   Introduction

The rapid progress in Text-to-image (T2I) generative models [29, 33, 28] has opened new creative avenues such as content generation [38, 3, 21] and editing [5, 39, 49, 22, 6, 48, 4], but users often desire customized outputs with specific topics or styles not present in the original training data [51]. Personalization techniques that fine-tune a pre-trained generative model on a small set of user-provided images have emerged to meet this need. Notably, methods like DreamBooth [30] and Low-Rank Adaptation (LoRA) [12] allow T2I models to be customized, capturing user-specific concepts (e.g. a particular pet, a unique face, or a distinct art style) and regenerating them in

39th Conference on Neural Information Processing Systems (NeurIPS 2025).

new contexts with high fidelity. While single-concept personalization is a relatively simple task, multi-concept generation is a challenging problem: Given multiple fine-tuned concept models (e.g. several LoRAs trained on different subjects), how can we compose them to synthesize a coherent image containing all the custom concepts? Achieving such compositions is challenging because independently trained LoRAs can interfere with each other when combined, leading to identity distortions or one concept dominating the other – a phenomenon sometimes called "LoRA crosstalk" [10, 25, 20, 36]. Simply merging or applying multiple LoRAs naively often causes one concept to vanish or entangle attributes with the other [10]. Recent research indeed highlights that multi-concept generation remains nontrivial: combining personalized models typically degrades individual concept quality unless special measures are taken [36, 25, 10]. However, these methods still require training a new combined model or a fine-tuning process (e.g., imposing constraints during each LoRA's training or running a post hoc alignment optimization).

While existing techniques can achieve multi-concept generation – i.e. producing a new image containing several personalized concepts – **none of these methods addresses the task of multi-concept editing**: modifying a given image to insert multiple new concepts. Multi-concept image editing presents a different set of challenges. Here, the goal is not to generate a scene from scratch, but to start from an input image and seamlessly blend in additional personalized elements (each defined by a LoRA model) into that image. A naive approach to this problem might be to apply iterative inpainting: for example, masking a region in the image and prompting the diffusion model (with the LoRA loaded) to generate the new concept in that area. Unfortunately, off-the-shelf inpainting with personalized diffusion models often yields artifacts and inconsistencies. The injected object or character may not blend naturally with the lighting and context of the original image, or the model may unintentionally alter the surrounding content. Another approach could be face-swapping or identity transfer, where a person's face in the image is replaced with a personalized face (using a LoRA of that person). Although this can handle a single face, it often does not preserve the full appearance of the person, such as body features, and can produce unrealistic results.

In this paper, we propose LoRAShop, a novel framework that enables multi-concept image editing with LoRA models, without requiring any additional training, special auxiliary inputs, or external segmentation. Given an input image and a set of LoRA modules (each encoding a different concept), LoRAShop allows the user to insert each concept into the image at a desired location in a disentangled way. One of our key observations is a disentangled mask extraction technique that leverages the internal representations of the rectified-flow model to localize the influence of each subject to be personalized. In essence, as each LoRA is applied during the denoising process, our method extracts a coarse mask that delineates the regions where that concept significantly contributes to the image. By combining these masks with the user's concept specifications, LoRAShop is able to blend multiple concepts directly into the diffusion latent in a controlled manner (see Fig. 1). Our experiments show that LoRA subjects blend naturally into the original scene, and their identities/styles match the LoRA concepts with high fidelity. Our approach does not require training of any new model or ensemble; it directly utilizes existing LoRAs and the base rectified-flow model at inference time, making it efficient and user-friendly. We believe that LoRAShop fills an important gap between personalized generation and image editing, opening the door to new creative workflows (such as "LoRAshopping" with generative models) that were previously impractical.

## 2 Related Work

**Personalized Image Generation.** Personalized image generation aims to inject a user-defined concept, typically a face, style, or object, into a text-to-image model so it can be used in future generations. Early work relied on Textual Inversion (TI) [9], which learns a single embedding that reproduces a user's concept. TI is lightweight, but struggles to learn concepts involve high level of detail, where it learns to reconstruct the target concept with diffusion loss. DreamBooth (DB) [30] improves fidelity by fine-tuning selected model weights and reserving a rare token for the new concept, though at a higher compute cost. Later methods seek better quality–efficiency trade-offs: $\mathcal{P}+$ [40] extends TI with a richer token representation; Custom Diffusion [19] trains only cross-attention layers; and DB-LoRA [32] applies low-rank adaptation [12] to store each concept in a small rank-limited update. Recent encoder-based systems such as StyleDrop [37], HyperDreamBooth [31], Taming Encoder [15], IP-Adapter [46], MS-Diffusion [42], MIP-Adapter[13], InfiniteYou [16], OmniGen

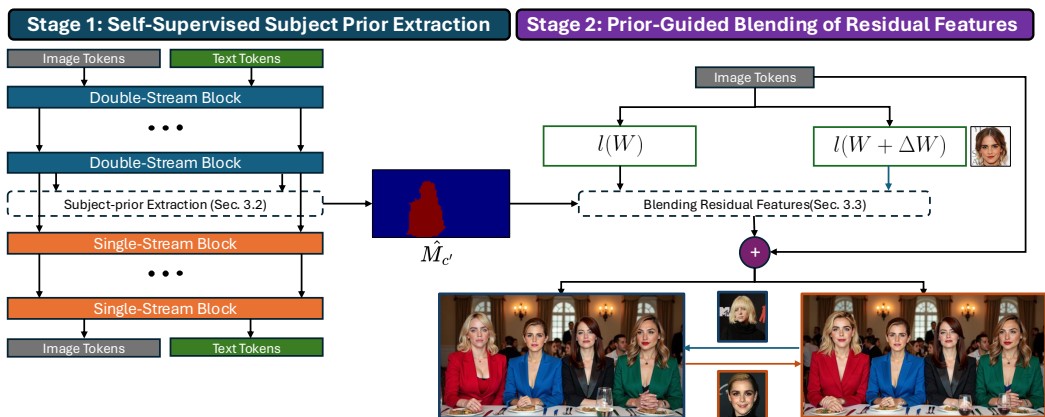

Figure 2: **LoRAShop Framework.** LoRAShop enables multi-subject generation and editing over a two-stage training-free pipeline. First, we extract the subject prior $\hat{M}_{c'}$, which gives a coarse-level prior on where the concept of interest, $c'$, is located. Following, we introduce a blending mechanism over the transformer block residuals, which both enables seamless blending of customized features and bounds the region-of-interest for the LoRA adapter utilized.

[45] and UNO [43] predict adapter features directly from reference images, enabling near-instant personalization but often with some loss of identity fidelity compared with full DreamBooth tuning.

**Merging Multiple Concepts.** Combining LoRAs for style and subject control remains as a challenging tasks, as combined adapters usually optimize overlapping representations. In achieving such a combination of personalized concepts, current work still faces certain challenges. Simple weight averaging [32] is fast but quickly causes interference. Mix-of-Show [10] trains special embedding-decomposed LoRAs that avoid this clash, yet it needs the original data and cannot use community models, such as those available on platforms like civit.ai [2]. ZipLoRA [34] merges one style and one content adapter but breaks down with more than one content LoRA. On the other hand, OMG [18] is based on an external segmenter to apply separate concepts, whose errors propagate to the result. Orthogonal Adaptation [25] keeps LoRAs in separate subspaces with additional constraints introduced, reducing cross-talk, but adds training overhead and likewise assumes data access. Our proposed approach differs from existing multi-concept generation methods since our main goal is 'editing' as opposed to generation. Moreover, our method does not require any input conditions such as keypoints or segmentation masks.

## 3 Method

We propose **LoRAShop**, a new training-free pipeline that enables the use of multiple LoRA adapters through a targeted feature blending scheme for multi-subject generation and editing. Our method, **Multi-Subject Residual Blending (MSRB)**, consists of two fundamental stages: 1) the extraction of a subject prior that effectively highlights the spatial regions where each subject is intended to appear, and 2) the application of a residual feature blending scheme within the diffusion transformer that selectively merges the outputs of different LoRA adapters. This allows us to spatially combine features corresponding to distinct concepts, enabling coherent and disentangled multi-subject generation and editing without any additional training.

### 3.1 Preliminaries

**Multi-Modal Diffusion Transformers.** Multi-modal diffusion transformers (MM-DiT) [8] extend the DiT architecture by processing text and image tokens in two tightly coupled streams, enabling end-to-end text-to-image generation. Rectified-flow models such as FLUX adopt this design and alternate between two transformer block types. We denote blocks that keep *separate* parameter sets for the text and image streams as *double-stream* blocks, and those that apply a *shared* transformation to both streams as *single-stream* blocks. During the denoising trajectory, the network first aligns textual and visual features within the double-stream blocks and subsequently refines the fused representation in

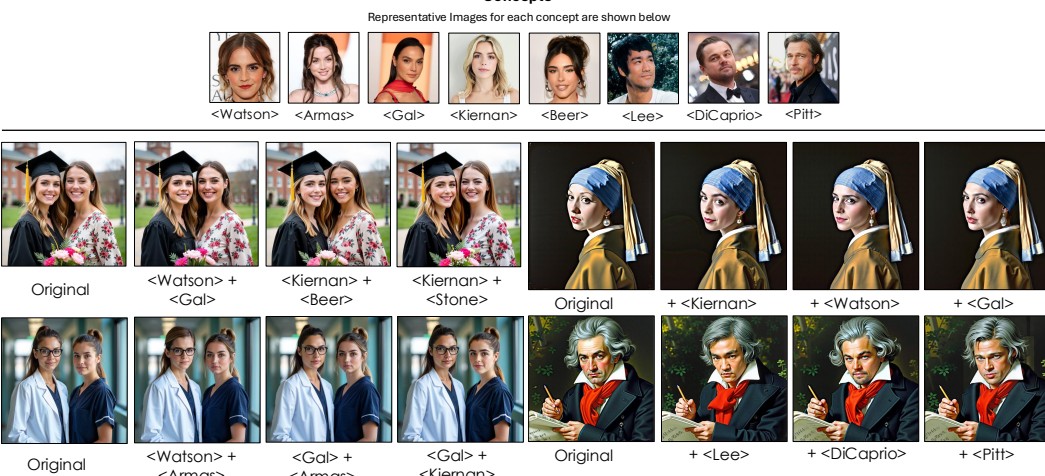

Figure 3: **Editing Generated & Real Images with LoRAShop.** We provide qualitative editing results with different human concepts. LoRAShop can achieve both edits on real and generated images. Due to non-intersecting subject prior extraction scheme of our framework, LoRAShop can perform edits with multiple concepts in one denoising pass.

the single-stream blocks. All feature updates propagate through residual connections, an architectural property that our generation and editing protocol leverages directly.

**Personalization via Low-Rank Adaptation.** Low-Rank Adaptation (LoRA) [12] was originally introduced as a lightweight fine-tuning method for large language models. Instead of updating the full weight matrix $W_0 \in \mathbb{R}^{d \times k}$, LoRA learns a low-rank increment, formulating the fine-tuned weights as $W = W_0 + \Delta W = W_0 + BA$ with $B \in \mathbb{R}^{d \times r}$, $A \in \mathbb{R}^{r \times k}$, and an *intrinsic rank* $r \ll \min(d, k)$. Because only $A$ and $B$ are trained, the additional parameter count and memory footprint scale linearly with $r$, making LoRA especially attractive for large backbones. Following its success in NLP, LoRA has been adopted for text-to-image diffusion models and, more recently, for rectified-flow transformers such as FLUX. We leverage single-subject LoRA adapters trained for rectified-flow transformers and introduce a training-free mechanism that allows multiple adapters, each corresponding to a different subject, to be used simultaneously without any additional optimization.

### 3.2 Self-Supervised Subject Prior Extraction

Training several LoRA adapters so they can be applied *simultaneously* is costly and often infeasible for large-scale denoisers: Every additional adapter consumes optimization memory, and jointly fine-tuning many of them tends to introduce interference and distribution drift. To bypass this bottleneck, we first predict, in inference time, where each personalized subject will emerge in latent space and then confine every adapter's effect to the pixels assigned to that subject. The binary masks that delimit these regions are our *subject priors*. We extract each prior once in a short *pseudo-denoising* run that proceeds only until timestep $\gamma$, when latents are still close to noise, yet cross-attention already carries strong spatial cues [11, 4]. Rectified flow transformers such as FLUX provide well-localized cross-attention maps. In particular, the map from the last block that still keeps the text and image streams separate (the *double-stream* block) gives the sharpest separation. For a prompt $c$ and the token subset $c'$ naming one subject we compute $M_{c'}$ where $Q_i$ are the image queries, $K_{c'}$ the keys of $c'$, and $d$ the key dimension.

$$M_{c'} = \text{softmax}\big(Q_i K_{c'}^{\mathsf{T}} / \sqrt{d}\big), \tag{1}$$

Because raw attention may fragment, we iteratively blur $M_{c'}$ with a $3 \times 3$ Gaussian kernel and renormalize until the super-threshold area forms a single connected component. Thresholding at the $\tau$ posterior quantile then produces the final binary mask, which we denote by $\hat{M}_{c'}$, for the subject $c'$.

When multiple subjects are present, these masks can intersect, leading to undesirable "LoRA cross-talk". To obtain non-overlapping maps, we stack the smoothed attention maps $\{\widetilde{M}_u\}_{u=1}^N$, determine for every spatial position $(i, j)$ the subject $u$ with the strongest response,

$$k^\star(i,j) = \arg\max_u, \widetilde{M}_u(i,j), \qquad \widetilde{M}_{\max}(i,j) = \widetilde{M}_{k^\star(i,j)}(i,j), \tag{2}$$

and finally define one-hot priors as $\hat{M}_u(i,j) = \mathbf{1}! \big[ u = k^\star(i,j) \big]$. The set $\hat{M}_u$ partitions the latent canvas without overlap and serves as the spatial guide for adapter mixing during generation and editing.

### 3.3 Prior-Guided Blending of Residual Features

The diffusion transformer proceeds as usual, but at every block we overwrite the *residual feature tensors* wherever a subject prior is active. At block $\ell$ the frozen backbone produces a collection of $R$ residual tensors, $\mathbf{F}_{\ell,r}^{\text{base}} \in \mathbb{R}^{S \times C}$ with $r = 1, \dots, R$, corresponding to the outputs of multi-modal attention, MLP, and any other sublayer that feeds a skip connection. In parallel, the $k$-th LoRA adapter contributes its counterparts $\mathbf{F}_{\ell,r}^{(k)}$. The binary priors $\hat{M}_{c'} \in \{0,1\}^S$ indicate which latent tokens belong to subject $c'$.

For each token position $p$ we turn the priors into weights, so the weights sum to one on the subject tokens and to zero on background tokens.

$$\alpha_{c'}(p) = \frac{\hat{M}_k(p)}{\sum_{u=1}^N \hat{M}_u(p) + \varepsilon}, \qquad \varepsilon \ll 1, \tag{3}$$

Whether the block is double-stream (text and image kept separate) or single-stream, we treat it the same way: *only image tokens are blended; prompt tokens keep their backbone residuals*. For every image token $p$ and every residual index $r$ we substitute

$$\tilde{\mathbf{F}}_{\ell,r}(p) = \sum_{k=1}^N \alpha_{c'}(p) \, \mathbf{F}_{\ell,r}^{(k)}(p), \tag{4}$$

and feed $\tilde{\mathbf{F}}_{\ell,r}$ back through the block's skip connection. If no subject claims token $p$ ($\sum_u \hat{M}_u(p) = 0$), we leave $\mathbf{F}_{\ell,r}^{\text{base}}(p)$ unchanged. Blending is disabled during the first until timestep $t$, letting the backbone establish the overall layout of the scene before subject-specific features are inserted. Because we mix the residual outputs of every sublayer rather than changing any weights, all adapters remain independent, and each subject influences exactly the tokens selected by its prior across the entire depth of the transformer.

### 3.4 Editing with LoRAShop

LoRAShop intervenes *only* in the feature space of a rectified flow transformer: it neither modifies the noise schedule nor alters any model weights. During the reverse diffusion process, we overwrite residual features solely at token positions indicated by the subject priors, leaving all other tokens unchanged. Because this operation is local and linear, the global denoising trajectory, and thus the overall scene layout, remains intact. The same mechanism integrates seamlessly with inversion. We adopt the RF-Solver pipeline of [41], which uses a second-order solver to recover the latent noise corresponding to a target image. After reconstructing the latent, we utilize LoRAShop to edit the inverted latent. As illustrated in Fig. 1 and Fig. 3, this enables region-controlled insertion of multiple personalized concepts into real images while faithfully preserving the properties of the input.

## 4 Experiments

We evaluate **LoRAShop** on both image generation and image editing tasks. For generation, we measure how well the method renders a single personalized subject and how reliably it composes multiple personalized subjects in one scene. For editing, we evaluate identity transfer on real images, replacing a person's appearance with that encoded by a LoRA adapter. We provide the details of our experimental protocol along with the results in this section.

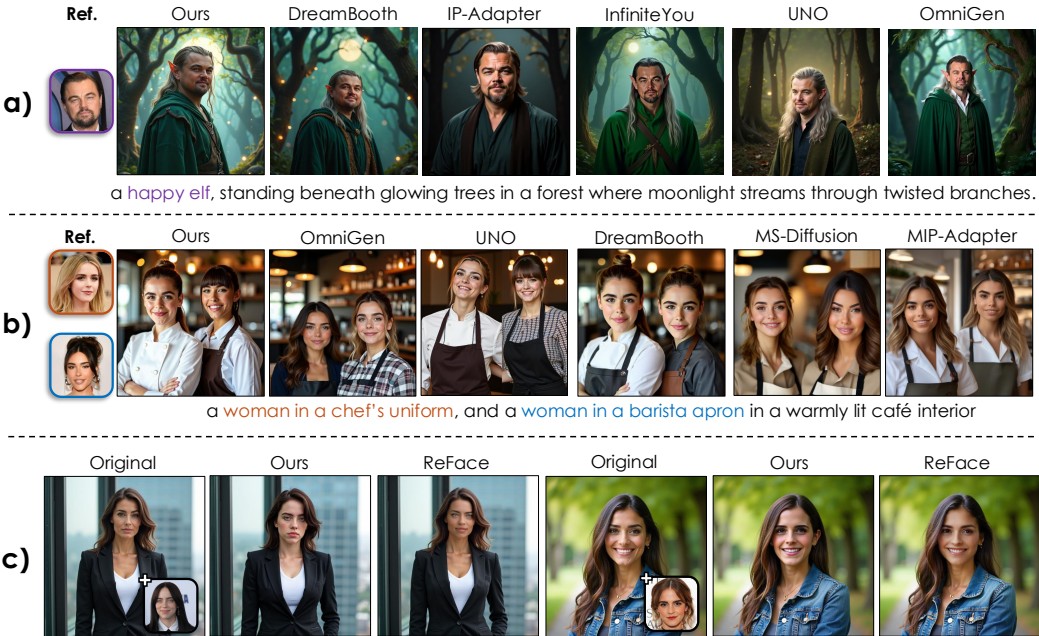

Figure 4: **Qualitative Comparisons.** We provide qualitative comparisons on three mainstream tasks: single-subject generation, multi-subject generation and face swapping. Over all of the benchmarked tasks, LoRAShop provides superior performance against competing approaches.

**Experimental Setup** We use `FLUX.1-dev`, as the rectified-flow transformer on which we build our approach. Our approach is based on utilizing pre-trained LoRA adapters for tasks such as single/multi-concept generation and editing. In all of our experiments, we use the LoRAs available at `diffusers` [24] library. We provide a complete list of LoRAs used in our experiments in the supplementary material, along with visual representations of these concepts for ease of understanding. Unless otherwise mentioned, we set the editing timestep $t = 0.90$, $\gamma = 0.94$ and $\tau = 0.7$, where we apply the proposed blending scheme (Sec. 3.3) onward timestep $t$ during the reverse process. Our approach requires no training over the pre-trained adapters and can perform the aforementioned personalization task in inference time. We conduct our experiments using one NVIDIA L40S GPU. LoRAShop can generate images using two concepts approximately in 50 seconds, as opposed to the manual inference time of `FLUX.1-dev` which requires 30 seconds per image. Furthermore, since LoRAShop can apply each concept sequentially, we introduce no memory constraints on how many concepts that can be applied to a given image. See Fig. 2 for 4 subject generation results.

Table 1: **Quantitative Comparisons on Single-Subject Generation.** We provide quantitative comparisons on single-subject generation. Our method outperforms the competing FLUX-based approaches in the overall performance, measured over identity similarity, prompt alignment and visual quality.

| Method | ID ↑ | CLIP-T ↑ | HPS ↑ | Aesthetics ↑ |
|---|---|---|---|---|
| DreamBooth | **0.755 ± 0.089** | 0.429 ± 0.055 | 0.305 ± 0.030 | 6.311 ± 0.505 |
| IP-Adapter-FLUX | 0.309 ± 0.077 | 0.330 ± 0.053 | 0.272 ± 0.026 | 6.340 ± 0.408 |
| InfiniteYou | 0.683 ± 0.068 | **0.439 ± 0.039** | 0.307 ± 0.026 | 6.490 ± 0.459 |
| Omni-Gen | 0.657 ± 0.066 | 0.434 ± 0.043 | 0.311 ± 0.030 | **6.514 ± 0.448** |
| UNO | 0.486 ± 0.137 | 0.415 ± 0.051 | 0.289 ± 0.030 | 6.303 ± 0.527 |
| **Ours** | 0.740 ± 0.066 | **0.439 ± 0.047** | **0.321 ± 0.028** | 6.499 ± 0.529 |

## 4.1 Qualitative Results

We qualitatively assess the effectiveness of our method in single & multiple subjects as well as generated & real images. To assess the visual performance of our framework, we demonstrate its capabilities in experiments on human subjects. Although LoRAShop can also perform edits on a

variety of types of subject, we perform our experiments on human subjects due to the high level of details such concepts involve, and their wide-usage in customization tasks. Since our method requires no fine-tuning for LoRA-adapters, we can use any adapter trained for our base model. Furthermore, since our approach does not focus on a specific type of residuals (e.g. attention layer outputs), but operates on the overall representation space, we can also use LoRAs with different ranks and different sets of fine-tuned parameters together.

**Editing on Generated and Real Images.** We provide editing results with LoRAShop on both male and female subjects, where these LoRAs are trained with different sets of combinations, which involve different sets of weights, ranks, and presence of a trigger word. Presented in Fig. 3, LoRAShop can both perform edits on real & generated images, without altering any subject-independent details. Note that, since LoRA adapters offer us a way to utilize the rich semantics in the weight space of the denoiser, our approach can also perform changes to the body of the edited subject (Fig. 3, row 1), which exceeds the limits of the face swapping task and provide us an advanced way of editing images with customized concepts. Furthermore, as our subject prior extraction algorithm provides non-intersecting masks, our approach facilitates performing multiple edits with distinct LoRA adapters in a single denoising pass.

**Qualitative Comparisons.** We provide qualitative comparisons of our approach with competing methods on single-subject, multi-subject and face swapping tasks. Since our proposed approach performs generation by editing, we enable blending of the residual features. While vanilla approaches such as DreamBooth [30] achieve subject-based generation results, since they fine-tune the weights of the original denoiser, they result in reduced prompt alignment and visual coherence. On the other end, encoder-based approaches such as IP-Adapter [46], InfiniteYou [16], UNO [43] and OmniGen [45] struggle to encode the identity features that are effectively captured by DreamBooth. In this regard, our approach offers the best of both worlds (Fig. 4 (a)), where we personalize only the regions related to the identity, which both achieves superior prompt alignment and personalization performance.

For multi-subject generation, we provide comparisons with FLUX-based approaches such as UNO [43], OmniGen [45], DreamBooth [30] and SDXL[26]-based approaches MS-Diffusion [42] and MIP-Adapter [13]. We use federated averaging for DreamBooth, as a baseline towards multi-subject personalization. As we demonstrate Fig. 4 (b), our subject priors mitigate the confusion between similar concepts effectively, where the remaining approaches either attempt to merge the two identities into one, or fail to capture the identity accurately. In this regard, our approach outperforms the competing methods for multi-subject generation as a training-free solution, which can effectively reflect multiple concepts effectively, mitigating the "cross-talk" effect between the concepts. Additionally, we provide comparisons with methods combining multiple LoRA adapters in Fig. 6, where our method offers compositions with high quality, without any pose input. To benchmark our method in terms of editing, we select the face swapping task, where we use identity LoRAs to represent the identity to be inserted into the original image. As we qualitatively benchmark in Fig. 4 (c), our approach extends the limitations of identity swapping, which was a task that is limited with swapping the faces until today. As LoRA adapters are capable of capturing physical features in addition to facial features, LoRAShop enables the transfer of physical features in addition to the face of the source identity, in addition to superior fidelity against methods based on inpainting such as ReFace [1].

While our main results focus on human subjects due to the complexity of identity fidelity, LoRAShop is not limited to face- or body-based personalization. In Fig. 6, we further demonstrate multi-concept composition on non-human and object-centric LoRAs (e.g., animal identities, accessories, stylized materials, and textured objects). These examples highlight that our residual blending operates on concept semantics rather than facial structure, and extend beyond identity transfer to general personalized content insertion. Quantitative evaluations for non-human concepts are provided in the supplementary material.

## 4.2   Quantitative Results

Extending our benchmark in qualitative experiments, we benchmark the editing and generation performance of LoRAShop on three mainstream tasks. Specifically, we benchmark the performance of LoRAShop for single & multi concept generation along with face swapping task. We provide the details of each constructed benchmark below.

Table 2: **Quantitative Comparisons on Multi-Subject Generation.** We benchmark our approach against FLUX and SDXL based methods. LoRAShop achieves superior identity preservation over multiple subjects, while maintaining the prompt alignment and visual quality of the base model.

| | Method | ID ↑ | CLIP-T ↑ | HPS ↑ | Aesthetics ↑ |
|---|---|---|---|---|---|
| **SDXL based** | OMG | $0.305 \pm 0.14$ | $0.217 \pm 0.09$ | $0.212 \pm 0.05$ | $6.017 \pm 0.35$ |
| | MS-Diffusion | $0.206 \pm 0.05$ | $0.251 \pm 0.08$ | $0.253 \pm 0.03$ | $6.119 \pm 0.24$ |
| | MIP-Adapter | $0.209 \pm 0.06$ | $0.243 \pm 0.07$ | $0.236 \pm 0.03$ | $6.111 \pm 0.30$ |
| **FLUX based** | DreamBooth | $0.444 \pm 0.08$ | $0.248 \pm 0.08$ | $\underline{0.259 \pm 0.04}$ | $6.113 \pm 0.30$ |
| | OmniGen | $\underline{0.453 \pm 0.09}$ | $\mathbf{0.256 \pm 0.08}$ | $0.258 \pm 0.04$ | $\mathbf{6.264 \pm 0.26}$ |
| | UNO | $0.270 \pm 0.07$ | $0.252 \pm 0.08$ | $0.255 \pm 0.04$ | $6.113 \pm 0.36$ |
| | **Ours** | $\mathbf{0.532 \pm 0.12}$ | $\underline{0.252 \pm 0.08}$ | $\mathbf{0.260 \pm 0.04}$ | $\underline{6.124 \pm 0.29}$ |

Table 3: **User Study.** We present user study results on identity preservation (Q1), and prompt alignment (Q2) for multi-subject generation task.

| | Method | User Study - Q1 ↑ | User Study - Q2 ↑ |
|---|---|---|---|
| **SDXL based** | OMG | $2.591 \pm 0.25$ | $3.332 \pm 0.55$ |
| | MS-Diffusion | $2.596 \pm 0.27$ | $2.753 \pm 0.19$ |
| | MIP-Adapter | $2.889 \pm 0.26$ | $3.123 \pm 0.50$ |
| **FLUX based** | DreamBooth | $3.196 \pm 0.10$ | $\underline{4.060 \pm 0.14}$ |
| | OmniGen | $\underline{3.340 \pm 0.32}$ | $4.012 \pm 0.26$ |
| | UNO | $2.711 \pm 0.23$ | $3.587 \pm 0.44$ |
| | **Ours** | $\mathbf{3.762 \pm 0.25}$ | $\mathbf{4.230 \pm 0.13}$ |

**Single-Subject Generation.** Following previous work, we populate a set of varying identities and generation prompts to benchmark our generation results. Among publicly available LoRA adapters, we select 15 identity LoRAs and generate a total of 520 images where 15 generation prompts were applied to each identity separately. To adequately assess both the personalization, prompt alignment and visual coherence of the generated outputs, we construct our benchmark prompts with themes such as artistic creations, contexts defined by activities and superficial concepts (see Fig. 4 (a)). We provide the complete list of prompts we use for our benchmark in the supplementary material. To assess both identity preservation, text alignment and visual coherence of the generated images, we utilize ArcFace embeddings [7], CLIP-T similarity [27], HPS score [44] and Aesthetics score[1]. We present the quantitative results in Table 1. As quantitative metrics also show, our approach leads to a sweet spot between identity preservation, prompt alignment, and visual coherence, as we utilize the generative priors in our residual blending scheme.

**Multi-Subject Generation.** In addition to our benchmark for single-subject generation, we also benchmark our approach against multi-subject generation methods. Using the 15 subjects that we used in our benchmark for single-subject generation, we initially generate random pairs of identities with corresponding prompts to create a benchmark for the two-subject generation task. In our evaluations, we compare our method with both FLUX-based methods UNO [43], OmniGen [45] and DreamBooth (FedAvg) [30] and SDXL-based methods OMG [18], MS-Diffusion [42] and MIP-Adapter [13]. As we present in the results in Table 2, our approach achieves both superior prompt alignment, visual coherence, and identity preservation.

**User Study.** Supplementary to our benchmark on multi-subject generation, we also conducted a user study to perceptually evaluate the generation quality of our approach. We conducted our study on 50 participants over `Prolific.com` crowdsourcing platform, where each participant is asked to assess 70 images involving multiple subjects. In our study, we evaluated the generation performance in which users are asked to rate the images in two aspects on a Likert scale (1: poor, 5: excellent): (Q1) alignment with the target identities and (Q2) alignment with the generation prompt. We provide the result of our study in Tab. 3. As our results also demonstrate, LoRAShop outperforms the competing

---

[1]`https://github.com/christophschuhmann/improved-aesthetic-predictor`

Table 4: **Quantitative Comparisons on Face Swapping.** We benchmark LoRAShop against REFace [1]. While performing on-par in input preservation, LoRAShop introduces significant improvements in identity preservation.

| Method | ID ↑ | DINO ↑ | CLIP-I ↑ | LPIPS ↓ |
|--------|------|--------|----------|---------|
| ReFace | $0.330 \pm 0.091$ | $\mathbf{0.982 \pm 0.012}$ | $\mathbf{0.940 \pm 0.038}$ | $\mathbf{0.031 \pm 0.033}$ |
| **Ours** | $\mathbf{0.709 \pm 0.101}$ | $0.970 \pm 0.019$ | $0.926 \pm 0.037$ | $0.050 \pm 0.019$ |

approaches in both prompt alignment and identity preservation. Please see Appendix for additional details about the user study.

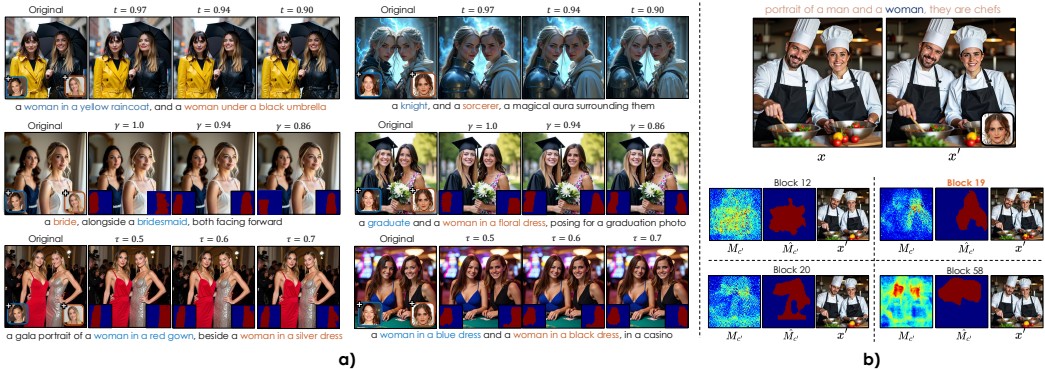

Figure 5: **Ablation Study.** (a) Ablations on hyperparameters time step $t$, subject's prior extraction step $\gamma$, and the posterior threshold for binarization of the subject's prior masks $\tau$. (b) Ablation on transformer blocks, where Block 19 shows superior ability for separation between subjects.

**Face Swapping.** We also benchmark our approach in the face swapping task. We compare our method with an inpainting-based swapping approach ReFace [1]. Although our approach does not involve any hard constraints for content preservation such as inpainting masks that restrict the regions to be edited, our method still achieves competitive performance in terms of input preservation, which we measure using DINO [23], CLIP-I [27], and LPIPS [50] metrics. Furthermore, LoRAShop leads to significant improvements in identity preservation properties. Note that our approach extends the bounds of the face swapping task and can perform full identity transfer by editing the physical appearance, in comparison to inpainting-based swapping approaches.

## 4.3 Ablation Studies

**Ablations on Transformer Blocks.** To further justify the use of the last double-stream block for subject prior extraction, and to provide an investigation over the roles of different transformer blocks, we provide ablations over the masks extracted from different transformer blocks in Fig. 5. As shown by the attention masks $M'_c$ extracted for the subject $c'$ (e.g. woman), we observe that through the double-stream blocks (blocks 0-19), FLUX constructs the semantic context and is able to perform the separation between different concepts at the end of these blocks. In the single-stream blocks, we observe that the model attempts to focus more on the visual details, which results in maps spread out over different entities. Building up on this observation, we build our subject prior extraction scheme on the attention maps produced by the last double-stream block (e.g. Block 19).

**Ablations on Editing Parameters.** Complementary to the block selection, LoRAShop includes three additional hyperparameters for editing, which are the editing time step $t$, the subject's prior extraction step $\gamma$, and the posterior threshold for binarization of the subject's prior masks $\tau$. We provide ablations on these hyperparameters in Fig. 5. Similarly to the trend observed in diffusion-based editing methods, LoRAShop is able to preserve the adapter-irrelevant features of the input image better when the edit is performed in later timesteps. Considering that the effect should be effective enough and preserve certain features of the input image, we achieve a good balance for the timestep $t$. Regarding the subject priors extracted prior to the denoising steps, we recognize that the introduced parameters have a significant impact on the quality of the mask. In general, we find $\gamma = 0.94$ and

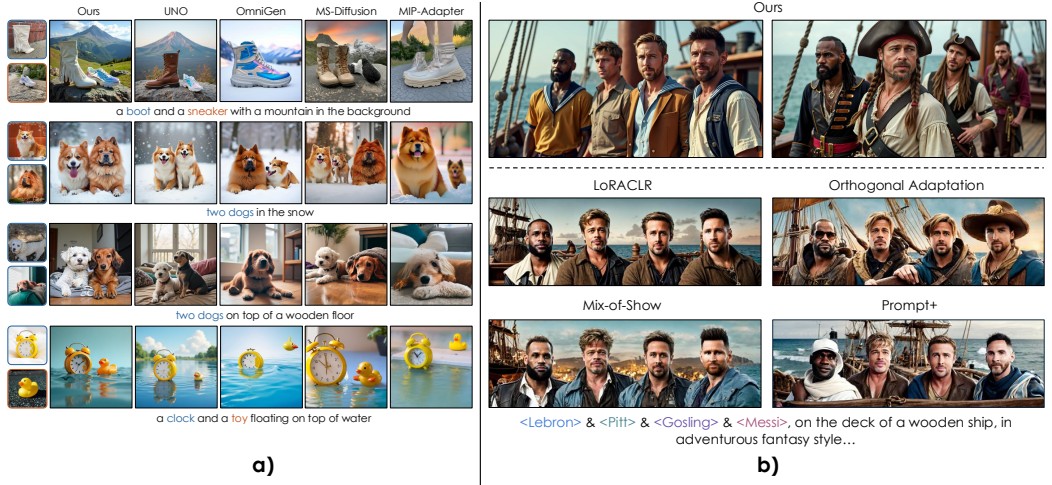

Figure 6: **Multi-concept personalization across domains.** **(a)** Non-human and object-centric composition: LoRAShop extends beyond identity personalization to animals, footwear, and objects, accurately preserving shape and texture semantics without retraining or external guidance. **(b)** Human multi-subject composition: LoRAShop disentangles and blends multiple identity LoRAs (e.g., *<Lebron>*, *<Pitt>*, *<Messi>*, *<Gosling>*), maintaining subject fidelity and spatial coherence. Compared to baselines; LoRACLR, Mix-of-Show, Prompt+, Orthogonal Adaptation, UNO, OmniGen, MS-Diffusion, and MIP-Adapter, LoRAShop preserves concept boundaries and contextual consistency while avoiding feature entanglement. Quantitative results for non-human concepts and occlusion-heavy overlapping cases are provided in the supplementary material.

$\tau = 0.7$ as suitable hyperparameters, which we utilize in all of our experiments for complete and accurate enough masks.

## 5 Discussion

**Limitations and Broader Impact.** Because the extracted masks inherit the latent biases of the underlying diffusion model (e.g., greater attention to faces, stereotypical gender features, or saturated colors) [17, 47], they can sometimes mislocate or underrepresent certain regions, leading to less coherent or unbalanced edits, particularly for concepts underrepresented in the model's pretraining data. Our mask extraction leverages attention patterns unique to the Flux architecture; other diffusion backbones (e.g., SDXL-Turbo) may require re-tuning of threshold parameters or yield less coherent masks. This limits immediate portability across all T2I models. Like other powerful editing tools, LoRAShop can be used to create non-consensual content. We encourage deployment within responsible-AI guardrails, but broader ethical safeguards remain necessary. Nevertheless, LoRAShop demonstrates—for the first time—training-free, region-controlled multi-concept editing with LoRAs, unlocking new creative workflows and research directions in compositional image manipulation.

**Conclusion.** We presented LoRAShop, the first training-free framework that enables region-controlled multi-concept image editing with off-the-shelf LoRA modules. By uncovering, and exploiting, spatially coherent activation patterns inside Flux diffusion transformers, we devised a disentangled latent-mask extraction procedure that lets each LoRA act only where it is intended, eliminating cross-concept interference. Without any extra optimization, segmentation, or auxiliary guidance, LoRAShop seamlessly blends multiple personalized subjects or styles into an input image, preserving both global context and fine local detail. Beyond advancing the state of the art in personalized image editing, LoRAShop turns diffusion models into an intuitive "photoshop-with-LoRAs," opening new possibilities for collaborative storytelling, product visualization, and rapid creative iteration.

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

# Table of Contents - Supplementary Material

# A    Details of User Study

We provide a sample question for the user study conducted in Fig. 7. To assess both the identity preservation and prompt alignment capabilities of our approach, we direct two questions to the participants of our study. The users are also provided representative examples of the personalized subjects, where these images are outsourced from assets available for public use. Then, the users are asked to rate the provided image on a Likert scale, where 1 corresponds to an unsuccessful generation and 5 corresponds to a successful generation.

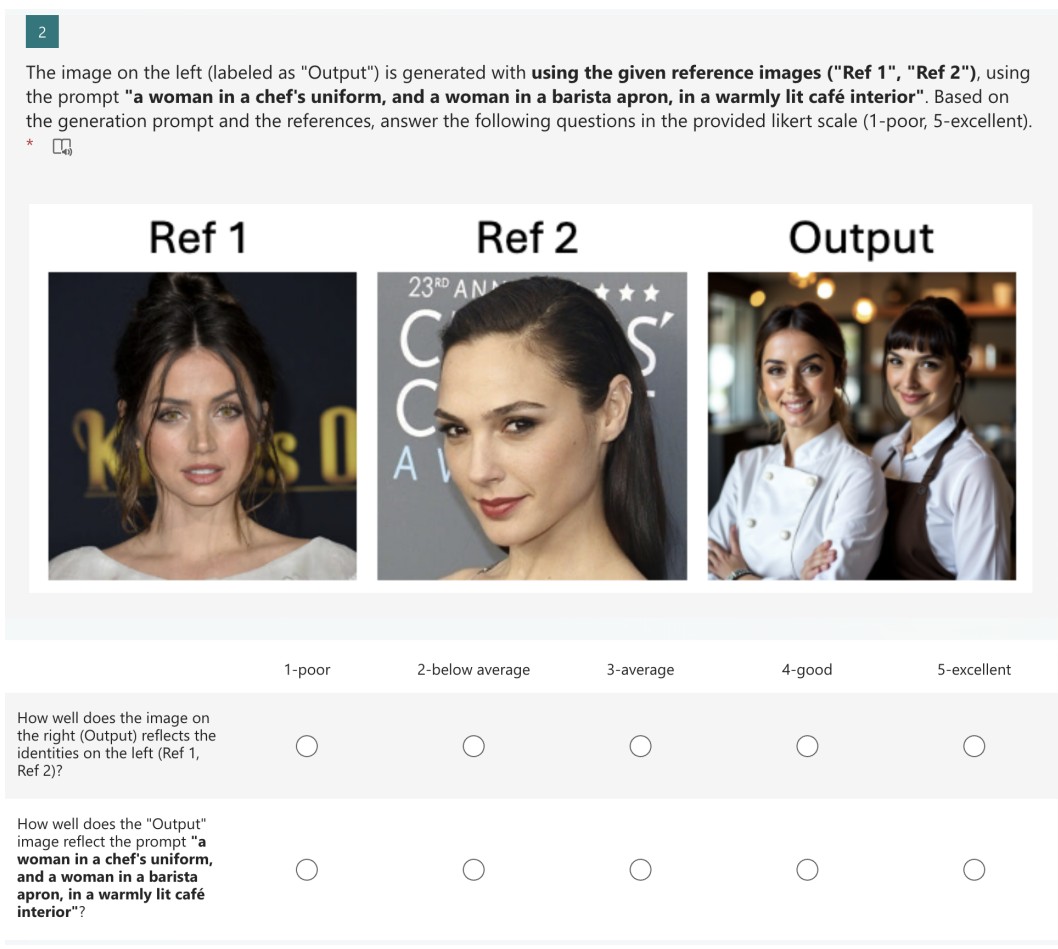

Figure 7: User Interface of our User Study.

# B    Additional Comparisons

## B.1    Comparisons with Multi-LoRA Composition Approaches

We compare our method against multi-concept LoRA composition approaches, including Mix-of-Show [10], LoRACLR [36], Orthogonal Adaptation [25], and Prompt+ [40] in Fig. 8. Notably, the first three methods require a pose condition for generating compositions, and the first two depend on specialized LoRA models such as EdLoRA, which limits their applicability when using community LoRAs from platforms like Civit.ai. In contrast, our method operates without pose conditions, retraining, or model merging, enabling successful composition using arbitrary LoRA models out of the box. Additionally, our method can compose LoRAs with different characteristics (e.g. different ranks and different sets of parameters), by operating on output space only. We also highlight that other methods do not support Flux, thus we visually compare with their Stable Diffusion-based generations.

We also note that LoRACLR [36], Orthogonal Adaptation [25], and Prompt+ [40] do not have publicly available implementations, which prevents us from conducting a quantitative comparison.

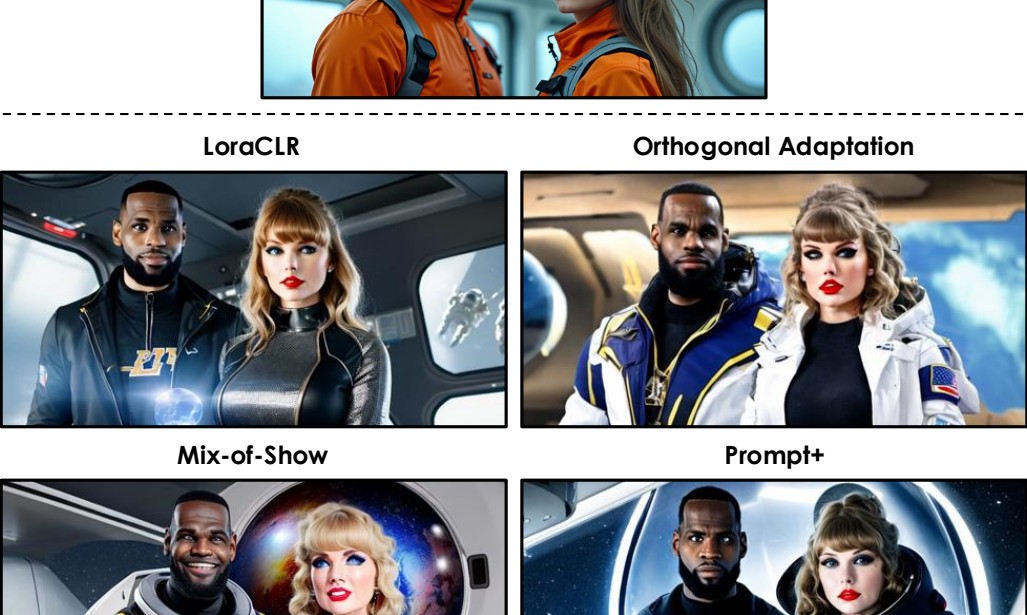

Figure 8: Comparison with state-of-the-art multi composition methods, on two subject generation task.

## B.2  Comparisons on the Extracted Masks

We additionally compare our subject-prior extraction strategy against ConceptAttention[11] masks under the same multi-subject prompt set used in the main paper. ConceptAttention does not reliably separate instances of similar concepts (e.g., two women), which leads to entangled subject regions during composition. In contrast, our subject priors are derived per-LoRA and remain instance-aware. We report the quantitative comparison in Table 5.

Table 5: **Quantitative comparison with ConceptAttention [11] on multi-subject scenarios.** We perform our comparisons using the same benchmark in the main paper. ConceptAttention fails to retain distinct identities when multiple similar concepts are present.

| Method | ID ↑ | CLIP-T ↑ | HPS ↑ | DINO ↑ |
|---|---|---|---|---|
| ConceptAttention | 0.362 | 0.250 | **6.189** | 0.261 |
| **Ours** | **0.532** | **0.252** | 6.124 | 0.261 |

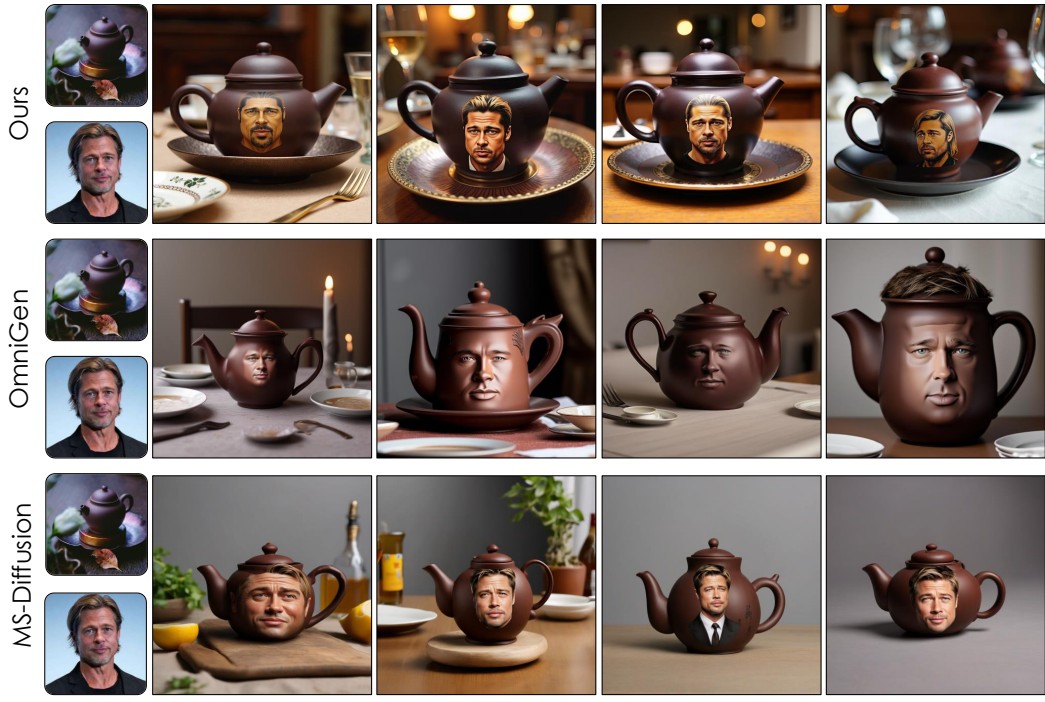

Figure 9: **Generations with occlusions.** When two concepts overlap spatially (e.g., a human identity and an object in the same region), competing approaches partially overwrite or distort one of the concepts. LoRAShop preserves both concepts in the final output.

## B.3 Quantitative Comparisons on Non-ID Concepts

We additionally evaluate LoRAShop on non-human concepts using the same multi-subject benchmark setup adopted in UNO. Specifically, we train 18 LoRAs using the concepts included in the DreamBooth dataset, form 10 concept pairs, and generate compositions using the standard prompt templates introduced by [30]. We report the quantitative comparisons in Table 6. As also reported in the quantitative evaluations, our method scores either best or second-best in all benchmarks.

Table 6: **Multi-subject comparisons with non-human concepts.** We additionally provide quantitative comparisons on multi-subject composition task with non-human concepts. For our evaluations, we train 18 LoRA adapters from the DreamBooth dataset and form 10 concept pairs.

| Method | CLIP-T ↑ | HPS ↑ | Aesthetics ↑ | DINO ↑ |
|---|---|---|---|---|
| MS-Diffusion | 0.483 | 0.291 | 5.609 | 0.446 |
| MIP-Adapter | 0.414 | 0.244 | 5.561 | 0.383 |
| OmniGen | 0.468 | 0.273 | **5.960** | 0.364 |
| UNO | **0.493** | 0.301 | **5.960** | 0.383 |
| **Ours** | 0.487 | **0.307** | 5.923 | **0.473** |

## B.4 Comparisons on Occluding Concepts

We also evaluate cases where two concepts occupy overlapping or partially occluded regions. In these scenarios, baselines often merge or partially overwrite one of the concepts, while our method preserves both concepts in the final composition. we provide qualitative examples in Fig. 9. Specifically, we use the LoRA adapter corresponding to concept *<Pitt>* and *<Teapot>* (from DreamBooth [30]) dataset. Additionally, we provide quantitative comparisons for this generation in Table 7. We use the prompt "a teapot with a face embossment on it, on a dinner table."

Table 7: **Quantitative Comparisons on Occlusion Cases.** To quantitatively evaluate the performance of our method on occlusion cases, we perform compositions with the concepts *<Pitt>* and *<Teapot>*. In our comparisons we perform 50 generations per method with the prompt "a teapot with a face embossment on it, on a dinner table". To measure the concept consistency, we report ID score for the human identity, and DINO score for the teapot.

| Method | ID ↑ | DINO ↑ | CLIP-T ↑ | HPS ↑ | Aesthetics ↑ |
|---|---|---|---|---|---|
| MS-Diffusion | 0.308 | **0.645** | 0.528 | 0.298 | 5.534 |
| OmniGen | 0.286 | 0.625 | 0.520 | 0.286 | 5.735 |
| **Ours** | **0.651** | **0.645** | **0.581** | **0.326** | **5.818** |

Table 8: **Full-body identity preservation user study.** We report pairwise user preferences for full-body identity preservation, where participants evaluate the identity transfer capabilities of competing methods under the same prompt set, where 1 corresponds to poor and 5 corresponds to excellent. LoRAShop is consistently preferred when the personalization extends beyond the face region to full body transfer.

| | OmniGen | InfiniteYou | Ours |
|---|---|---|---|
| **User Preference** | 2.608 | 2.848 | **3.228** |

## B.5 Evaluations on Full Body Identity Transfer

In addition to face-centric identity transfer, we also conducted a user study on 40 personalized samples to evaluate full-body identity preservation. We follow the same pairwise preference protocol as our main user study: participants are shown two images side-by-side and asked which one better matches the target identity for a given prompt, while specifying that the task is full-body identity preservation. We provide the results of our user study in Table 8, where we compare our method with OmniGen [45] and InfiniteYou [16].

The results indicate that LoRAShop is preferred over the competing approaches in the majority of comparisons, demonstrating that the extracted subject priors enable identity transfer at a whole-body level, not only at the facial region. These findings align with our qualitative results and confirm that the method generalizes to fine-grained, full-body personalization rather than relying solely on facial matching signals such as ArcFace.

## C   Supplementary Generation and Editing Examples

Supplementary to the editing and generation examples provided in the main paper, we provide supplementary results from *LoRAShop* in this section. Specifically, we provide examples of four subject generation in Fig. 10, three subject generation in Fig. 11, two subject generation in 12, and a combination of human and non-human adapters in Fig. 13. As we demonstrate qualitatively, our approach can both handle multiple instances of the same type of entities (e.g. woman) and different type of entities (e.g. man, sunglasses, clothing).

## D   Detailed Masking and Blending Algorithm

To further clarify the details of our method, we provide detailed descriptions of subject prior extraction and residual blending scheme introduced in the main paper. For the subject prior extraction, we provide the details of blob construction algorithm in 1. In addition, to further clarify the blending process, we describe the blending process for a given residual (from a transformer block) in Alg. 2. Note that this blending operation is applicable for all residual features outputed inside the transformer blocks.

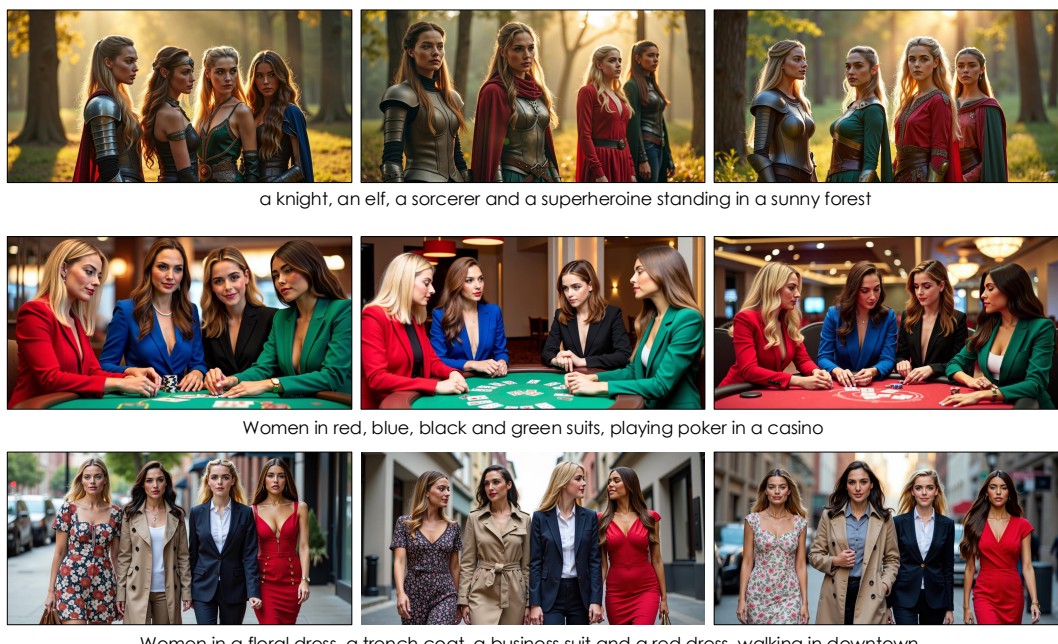

a knight, an elf, a sorcerer and a superheroine standing in a sunny forest

Women in red, blue, black and green suits, playing poker in a casino

Women in a floral dress, a trench coat, a business suit and a red dress, walking in downtown

Figure 10: Multi-subject composition results on four human subjects. As our approach does not rely on any other external conditioning like pose conditioning, *LoRAShop* can utilize the generative capabilities of FLUX, and thus generate outputs with high fidelity and superior prompt alignment. In the provided examples, we utilize the concepts `<Margot>`, `<Gal>`, `<Kiernan>` and `<Beer>`.

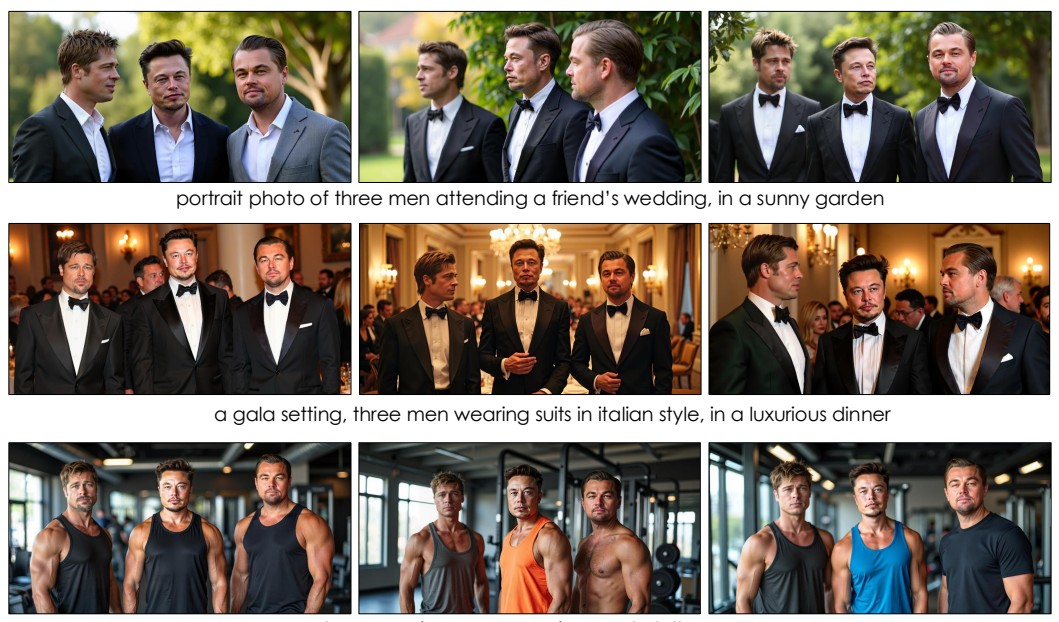

portrait photo of three men attending a friend's wedding, in a sunny garden

a gala setting, three men wearing suits in italian style, in a luxurious dinner

Three men in a gym, wearing sport clothes

Figure 11: Multi-subject composition results for three subjects. We provide generation results for the subjects `<Pitt>`, `<Elon>` and `<DiCaprio>`. We provide generation results on three different generation prompts, with different compositions of the subjects.

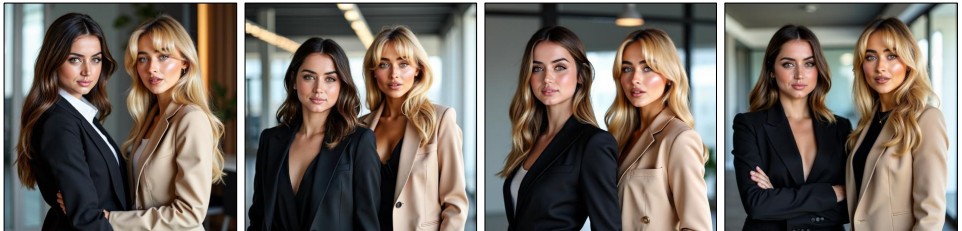

a woman in a black business suit, next to a woman in a beige blazer, in a modern office setting

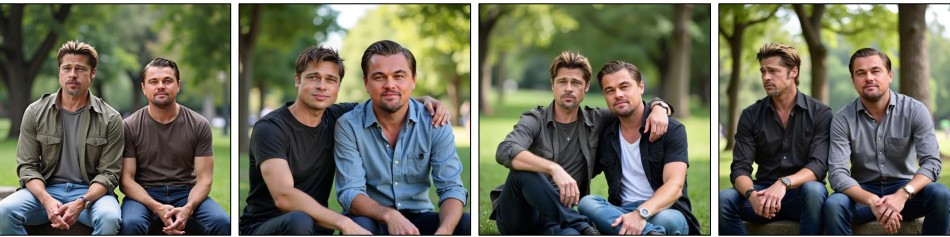

two men sitting in a park, wearing casual clothes

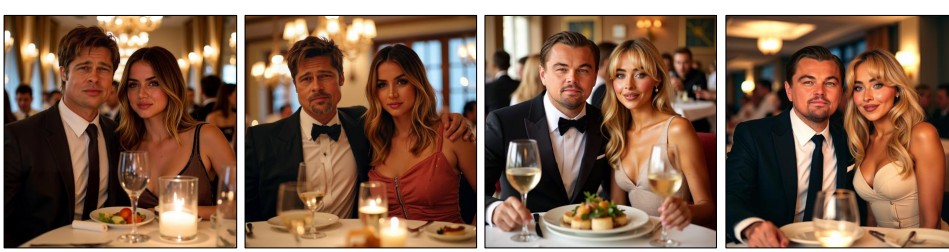

a man and a woman, having a business dinner in a fancy restaurant, posing for a photo

Figure 12: Multi-subject composition results for two subjects. We provide generation results for the concepts `<Armas>`, `<Sabrina>`, `<Pitt>`, `<DiCaprio>`. As we demonstrate in the examples, *LoRAShop* can perform compositions between the same type (e.g. woman-woman) and different type (e.g. man-woman) of identity concepts.

# E  Experiment Details

In this section, we provide supplementary details on our quantitative evaluations and provide the specifics of the metrics utilized and the prompt sets used. We provide the prompts that we use for the evaluation of the single-subject and multi-subject generation tasks in Table 12 and Table 13, where we generate the prompt set with GPT-4o[14]. In the following, we provide the details for each of the metrics that we use in our evaluations.

- **ID:** We use the `InsightFace`[2] codebase for the ID similarity metric. Specifically, we use ArcFace [7] embeddings provided in their implementation, using the `buffalo_l` variant.

- **CLIP:** To assess text-to-image similarity for single/multi subject generation tasks, and image-to-image similarity for face swapping benchmark, we utilize CLIP [27] as our feature extractor. In all of our experiments, we use the `big-G` variant of the model[3].

- **HPS:** As a secondary metric to quantify text-to-image alignment, we utilize the Human Preference Score (HPS), which is fine-tuned with user preferences. In our experiments, we use the `HPSv2`[4] variant.

---

[2]`https://github.com/deepinsight/insightface`
[3]`https://huggingface.co/laion/CLIP-ViT-bigG-14-laion2B-39B-b160k`
[4]`https://github.com/tgxs002/HPSv2`

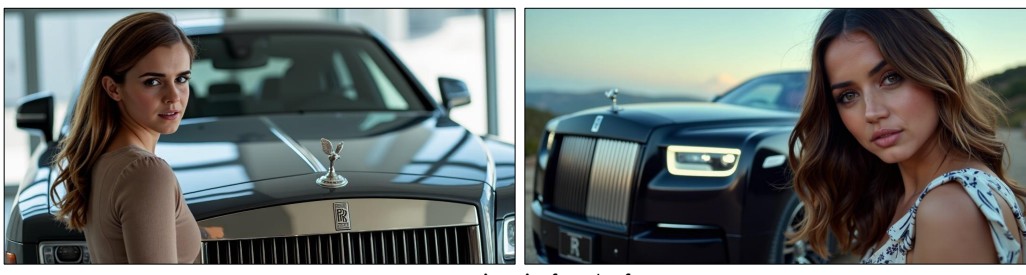

a woman posing in front of a car

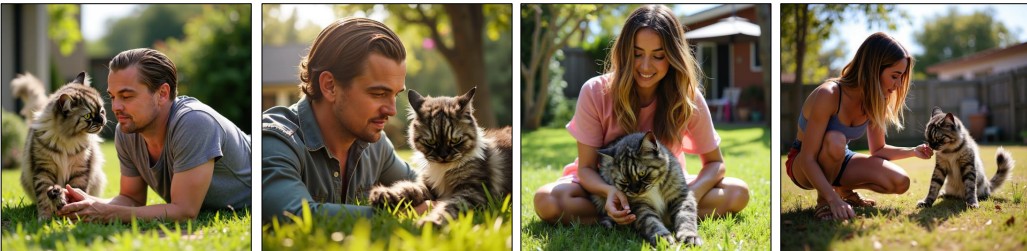

a man/woman and a cat, playing together in a sunny backyard

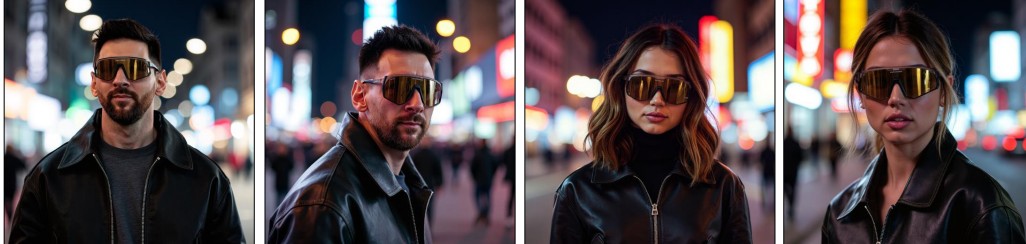

a man/woman with sunglasses, wearing a jacket. Standing in the downtown at night

Figure 13: Multi-subject composition results generated by our method on different types of objects. As can be seen in the examples *LoRAShop* can perform combinations between different types of concepts.

- **Aesthetics:** To assess the quality of the generated images, we utilize the aesthetics score for single and multi subject generation tasks. We use the second version of the predictor in all of our experiments[5].
- **DINO:** As a secondary metric to assess the input preservation for the face swapping task, we use DINO for our benchmark. We use the checkpoints from `https://huggingface.co/facebook/dinov2-base`.
- **LPIPS:** Following the common practice from image editing tasks, we utilize LPIPS [50] score with VGG [35] backbone.

For all of the competing approaches, we use the default hyperparameter setups and their corresponding official implementations.

## F   List of LoRA Adapters

We provide a complete list of LoRA adapters used in this section. Specifically, we provide the list of the LoRA adapters for woman subjects in Tab. 9, man subjects in Tab. 10 and non-human subjects in Tab. 11. For each of the adapters, we provide representative images for each, to help readers identify the subjects. Note that, we provide this list as a legend, where the adpater icons are in match with the ones used in the main paper. As exceptions, we train the LoRA adapters for `<Gosling>` and `<Lebron>` using Dreambooth [30].

---

[5] `https://huggingface.co/shunk031/aesthetics-predictor-v2-sac-logos-ava1-l14-linearMSE`

| Original | +<Watson> | +<Gal> | +<Kiernan> | +<Beer> |
|----------|-----------|--------|------------|---------|

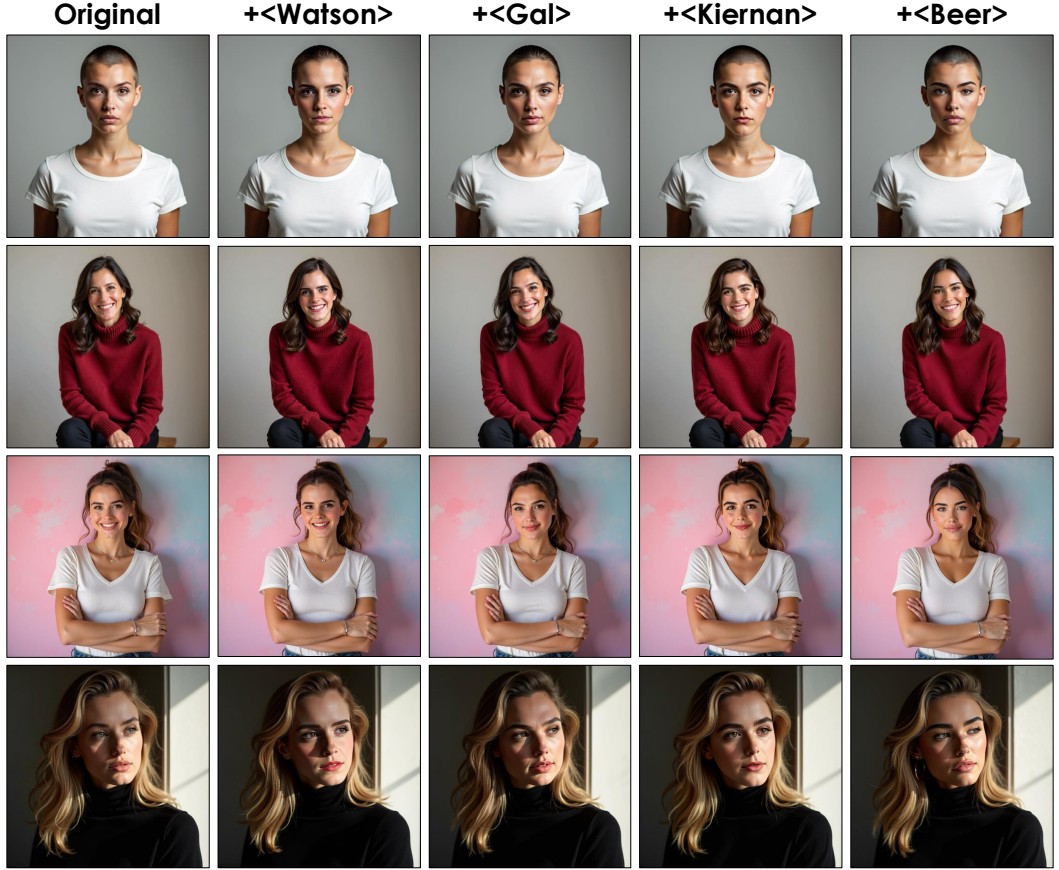

Figure 14: Face Swapping results with *LoRAShop*. As we demonstrate in the provided examples, our editing approach offers a seamless blending between the input subject and the target identity, while preserving the input characteristics.

| Original | +<Watson> | +<Gal> | +<Kiernan> | +<Beer> |

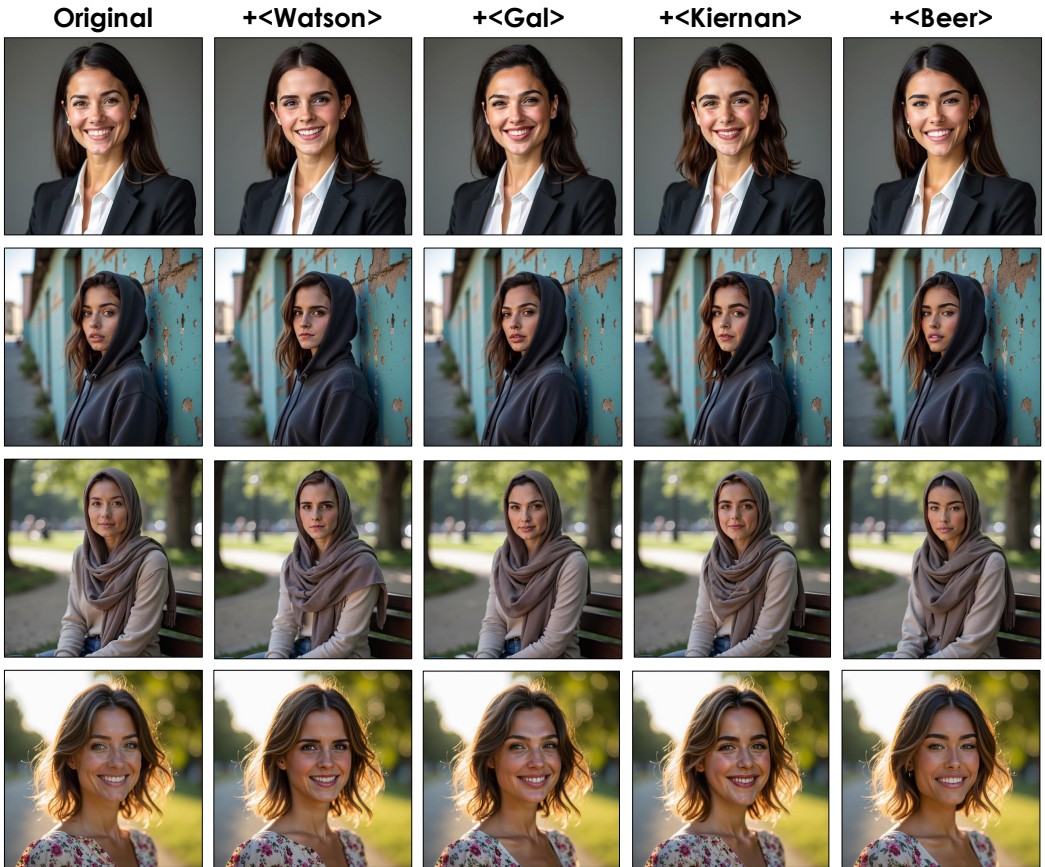

Figure 15: Face Swapping results with *LoRAShop*.

---

**Algorithm 1** HOMOGENEOUSBLOB

---

**Require:** Soft mask $\mathbf{M} \in [0,1]^{B \times H*W \times 1}$, image size $(H, W)$, Gaussian size $k$, variance $\sigma$, threshold
    $t$, maximum passes $P$, mode `flatten`, distance parameter $\lambda$
1:  $\mathbf{M} \leftarrow \text{reshape}(\mathbf{M}, \langle B, 1, H, W \rangle)$
2:  $G \leftarrow \text{GaussianKernel}(k, \sigma)$
3:  $\mathbf{M} \leftarrow \text{renorm}(\mathbf{M})$                                         ▷ 0–1 scaling
4:  **for** $p = 1$ **to** $P$ **do**
5:      $\mathbf{M} \leftarrow \text{renorm}\big(\text{conv2d}(\mathbf{M}, G)\big)$
6:      **if** every batch sample has $\leq 1$ connected component above $t$ **then**
7:          **break**
8:      **end if**
9:  **end for**
10:                                           ▷ **Homogenise the blob**
11:  $\mathbf{P} \leftarrow \mathbb{1}_{\{\mathbf{M}=\max(\mathbf{M})\}}$               ▷ Use the global peak as a single-pixel marker
12:  $\mathbf{M} \leftarrow \text{morph\_reconstruct}(\mathbf{P}, \mathbf{M})$   ▷ Flood-fill outward until original mask intensity is reached
13:  $\mathbf{M} \leftarrow \text{renorm}(\mathbf{M})$              ▷ Rescale result to $[0, 1]$; yields a flat, uniform blob
14:  $\hat{\mathbf{M}} \leftarrow \text{reshape}(\mathbf{M}, \langle B, H*W, 1 \rangle)$
15:  **return** $\hat{\mathbf{M}}$

---

---

**Algorithm 2** RESIDUALBLENDING

---

**Require:**
  $\mathbf{R}^{\text{base}} \in \mathbb{R}^{S \times C}$                                                      ▷ residual from frozen backbone
  $\mathbf{R}^{(k)} \in \mathbb{R}^{S \times C}$ for $k = 1, \ldots, N$                                 ▷ residuals from $N$ LoRA adapters
  $\hat{M}_k \in \{0, 1\}^S$ for $k = 1, \ldots, N$                                    ▷ token-wise subject priors
  $\mathcal{I} \subseteq \{1, \ldots, S\}$                                                 ▷ indices of image tokens
  $\varepsilon$ small constant                                               ▷ avoids divide-by-zero
**Ensure:** $\widetilde{\mathbf{R}} \in \mathbb{R}^{S \times C}$                                      ▷ blended residual tensor
 1: **for each** token index $p = 1, \ldots, S$ **do**
 2:     **if** $p \notin \mathcal{I}$ **then**                                       ▷ prompt token: no blending
 3:         $\widetilde{\mathbf{R}}(p) \leftarrow \mathbf{R}^{\text{base}}(p)$
 4:         **continue**
 5:     **end if**
 6:     sumMask $\leftarrow \sum_{u=1}^{N} \hat{M}_u(p)$
 7:     **if** sumMask $= 0$ **then**                                  ▷ background token
 8:         $\widetilde{\mathbf{R}}(p) \leftarrow \mathbf{R}^{\text{base}}(p)$
 9:     **else**                                           ▷ token claimed by a subject
10:         **for** $k = 1$ **to** $N$ **do**
11:             $\alpha_k \leftarrow \hat{M}_k(p)/(\text{sumMask} + \varepsilon)$                ▷ normalise prior to a weight
12:         **end for**
13:         $\widetilde{\mathbf{R}}(p) \leftarrow \sum_{k=1}^{N} \alpha_k \mathbf{R}^{(k)}(p)$      ▷ blend adapter residuals according to weights
14:     **end if**
15: **end for**
16: **return** $\widetilde{\mathbf{R}}$                                       ▷ ready for the block's skip connection

---

| Adapter Icon | Adapter Tag | URL of the Adapter |
|---|---|---|
|  | `<Armas>` | `https://huggingface.co/Trenddwdw/Ana_de_Armas` |
|  | `<Billie>` | `https://huggingface.co/punzel/flux_billie_eilish` |
|  | `<Watson>` | `https://huggingface.co/punzel/flux_emma_watson` |
|  | `<Gal>` | `https://huggingface.co/punzel/flux_gal_gadot` |
|  | `<Kiernan>` | `https://huggingface.co/punzel/flux_kiernan_shipka` |
|  | `<Margot>` | `https://huggingface.co/punzel/flux_margot_robbie` |
|  | `<Margot>` | `https://huggingface.co/punzel/flux_emma_stone` |
|  | `<Beer>` | `https://huggingface.co/punzel/flux_madison_beer` |
|  | `<Sabrina>` | `https://huggingface.co/mmaluchnick/sabrina-carpenter-flux-model` |
|  | `<Taylor>` | `https://huggingface.co/DeZoomer/TaylorSwift-FluxLora` |

Table 9: Image-and-text comparison table.

| Adapter Icon | Adapter Tag | URL of the Adapter |
|---|---|---|
|  | `<DiCaprio>` | `https://huggingface.co/openfree/`
`leonardo-dicaprio` |
|  | `<Pitt>` | `https://huggingface.co/Trenddwdw/Brad_Pitt` |
|  | `<Lee>` | `https://huggingface.co/openfree/bruce-lee` |
|  | `<Elon>` | `https://huggingface.co/roelfrenkema/flux1.`
`lora.elonmusk` |
|  | `<Messi>` | `https://huggingface.co/namita2991/messi` |

Table 10: Image-and-text comparison table.

| Adapter Icon | Adapter Tag | URL of the Adapter |
|---|---|---|
|  | `<Lumiva>` | `https://huggingface.co/Litqecko/`
`lumiva-glasses` |
|  | `<Jacket>` | `https://huggingface.co/Oscar2384/Loewe_`
`Hybrid_bomber_jacket_in_nappa` |
|  | `<Dress>` | `https://huggingface.co/martintomov/`
`moncler-dress-1000-v1` |
|  | `<Tower>` | `https://huggingface.co/seawolf2357/ntower` |
|  | `<Cat>` | `https://huggingface.co/ginipick/`
`flux-lora-eric-cat` |
|  | `<Royce>` | `https://huggingface.co/seawolf2357/`
`flux-lora-car-rolls-royce` |

Table 11: Image-and-text comparison table.

| ID | Prompt |
|----|--------|
| P 1 | a woman/man rendered in a stylized manner is centered in the image, standing in front of a backdrop of expressive brushstrokes and vibrant color blocks. |
| P 2 | a woman/man illustrated in pencil is centered in the frame, with fine shading and linework defining her/his face, placed against a softly sketched background. |
| P 3 | a woman/man illustrated with smooth digital brushwork is centered in the image, with soft ambient lighting and a clean gradient background behind her/him. |
| P 4 | a woman/man rendered in art deco style is centered in the scene, framed by angular gold patterns and symmetrical borders in an ornate composition. |
| P 5 | a woman/man is centered in the image, rendered in a cyberpunk painting style with neon reflections casting pink and blue highlights across her/his face, glowing circuitry traced along her/his cheekbones, and a blurred futuristic cityscape of holograms and rain-soaked signs behind her/him. |
| P 6 | a woman/man with a happy expression, sitting near a tall window with natural light falling across her/his face, while shadows from nearby plants frame the soft background. |
| P 7 | a beautiful woman/man is centered in a cozy room filled with bookshelves and warm lighting, her/his face lit by a glowing screen as she/he laughs during a video call. |
| P 8 | a woman/man with a nervous expression on a misty morning trail, the background gently blurs into distant trees and dew-covered grass. |
| P 9 | a woman/man with a happy expression in a warmly lit kitchen, preparing a meal with a relaxed expression, surrounded by ingredients and subtle reflections from the counter. |
| P 10 | a woman/man is centered at a café table, sketching in a notebook with soft light falling on her/his face, as the background softly fades into rustic textures and furniture. |
| P 11 | a woman/man knight with a fierce expression, wearing intricately detailed medieval armor, standing on a battlefield at sunset as orange light reflects off her/his head and the silhouettes of fallen weapons surround her/him. |
| P 12 | a woman/man sorcerer is centered in the image, casting a glowing spell with both hands, her/his face illuminated by swirling magical energy, while runes float in the air and a faint aura pulses around her/him in the twilight mist. |
| P 13 | a futuristic cyborg woman/man is centered in the image, with a metallic faceplate, cybernetic implants across her/his jaw and temple, and glowing blue circuitry along her/his neck, standing in front of a neon-lit skyline under a starless night sky. |
| P 14 | a woman/man dragon rider is centered in the image, her/his face framed by windswept hair and a dark leather hood, with the neck of a black-scaled dragon behind her/him and storm clouds swirling in the sky, her/his expression fierce and focused as wind lifts her/his cloak around her/his shoulders. |
| P 15 | a happy woman/man elf in a portrait photo setting, with long silver hair and pointed ears, cloaked in forest-green robes, standing beneath ancient glowing trees in an enchanted forest where magical particles float in the air and moonlight streams through twisted branches. |

Table 12: Prompt list for single-subject generation.

| ID | Prompt |
|---|---|
| P 1 | a close-up profile photo of a woman in a red suit with slicked-back hair and defined brows, next to a woman in a green suit with soft curls and a warm smile, both standing side by side under office hallway lighting, posing to the camera. |
| P 2 | a headshot-style image of a woman in a white lab coat with glasses and a sharp jawline, beside a woman in navy scrubs with tied-back hair and a round face, both facing forward in a hospital corridor. |
| P 3 | a portrait-style image of a woman in a floral dress with curly blonde hair and bright eyes, and a woman in a denim jacket with straight black hair and a neutral expression, both seated on a park bench, looking at the camera. |
| P 4 | a profile photo of a woman in a black business suit with a confident expression, next to a woman in a beige blazer with a composed look, both looking directly at the camera in a modern office setting. |
| P 5 | a woman in a crisp chef's uniform with her hair neatly tied back and a confident expression, and a woman in a barista apron with short bangs and a friendly smile, both posed for professional headshots in a warmly lit café interior. |
| P 6 | a head-and-shoulders photo of a woman in athletic wear with her hair tied up and a serious look, next to a woman in a hoodie with loose strands and a light smile, both standing on a track field at sunrise. |
| P 7 | a portrait-style image of a woman in a yellow raincoat with damp bangs and a composed face, and a woman under a black umbrella coat with a cheerful smile, both captured walking side by side on a rainy city street. |
| P 8 | a softly lit bridal portrait of a bride in a white wedding dress with glowing makeup, alongside a bridesmaid in a navy gown with a calm expression, both facing forward in a bridal room setting. |
| P 9 | a posed gala portrait of a woman in a red evening gown with defined features and dramatic makeup, beside a woman in a silver sequin dress with soft curls and a neutral expression, both under spotlight lighting. |
| P 10 | a construction site ID photo of a woman in a safety vest and hard hat with a firm gaze, next to a woman holding blueprints with glasses and a composed face, both framed in the foreground. |
| P 11 | a portrait of a woman holding a camera in casual wear with a focused look, and a woman with a soft gaze in a long white dress, both photographed at golden hour in a field. |
| P 12 | a greenhouse portrait of a woman in a green apron with tied-back hair and a relaxed expression, next to a woman in a plaid shirt with a gentle smile, both facing the camera with greenery in the background. |
| P 13 | a coastal roadside profile photo of a woman in a black motorcycle jacket with bold lipstick, and a woman in a sundress holding an ice cream cone with a cheerful expression, both posing beside a scooter. |
| P 14 | a woman with a calm expression sits at a café table, her face softly lit and clearly framed in the image; and a woman beside her with a gentle smile turns slightly toward her, both captured from the shoulders up in a warm, relaxed atmosphere with the background softly out of focus. |
| P 15 | a college campus profile photo of a graduate woman in a black gown and cap with a proud smile, and a woman in a floral dress holding a bouquet with a joyful expression, posing for a graduation photo. |

Table 13: Multi-subject prompts used in our evaluation.

