# OpenReview forum: "LoRAShop: Training-Free Multi-Concept Image Generation and Editing with Rectified Flow Transformers"
_NeurIPS.cc/2025/Conference — NeurIPS 2025 spotlight_

### Official Review · Reviewer_ddJF · 2025-07-01

**Clarity:** 3
**Significance:** 3
**Originality:** 2
**Rating:** 5
**Confidence:** 4

**Summary:**

To enhance the consistency of multiple IDs in image-to-image generation and mitigate the coupling effect, the author proposes leveraging the inherent semantic advantages of DiT—specifically, using cross-attention to first obtain masks for each ID to bind the corresponding LoRA. This is a simple yet effective approach, and its extension to other tasks demonstrates the robustness of the method.

**Questions:**

1. Why must it be the LoRA structure? Have other options been tried?
2. It feels like the test data is insufficient—only 15 identity LoRAs. We would prefer to see more diverse results.
3. The authors conducted ablation experiments on the mask extraction effect across different layers. Has something like ConceptAttention been attempted? Their method appears to be more robust.

**Ethical Concerns:**

["NO or VERY MINOR ethics concerns only"]

**Final Justification:**

The author explains the necessity of using the LoRA structure and provides additional experiments for more test scenes and different mask strategies. He has addressed my concerns, so I have adjusted my score to 5.

**Limitations:**

My primary concerns are twofold:
1) The paper predominantly focuses on facial features, leaving the effectiveness on other categories unclear;
2) Additional mask acquisition approaches should be explored, similar to ConceptAttention.

**Paper Formatting Concerns:**

No concerns about this.

**Quality:**

3

**Strengths And Weaknesses:**

Strengths: The system can conveniently leverage the abundant LoRA components within the community, demonstrating strong extensibility.

Weaknesses: In the quantitative results, the authors only employed identity LoRA, and the qualitative experiments primarily focused on facial images. The effectiveness on other categories remains unclear.

---

> ### Author Rebuttal · Authors · 2025-07-31
>
> We thank reviewer ddJF for the valuable comments and finding our method as strongly extensible and beneficial for the community. Please find our responses to the concerns raised below.
>
> **Q1: Why must it be the LoRA structure? Have other options been tried?**
>
> We chose to use LoRA adapters for two key reasons. First, LoRAs have become the de-facto standard for single-concept personalization, resulting in a large public pool of pretrained adapters that LoRAShop can readily leverage, thus enhancing the method’s practical utility and community impact. Second, prior studies such as Concept Sliders [1], HyperDreamBooth [2], and Weights2Weights [3] have shown that diffusion models organize semantic information effectively in weight space. By operating in this space through LoRA adapters, our method is able to capture full-body semantic features. While we did not experiment with alternative adapter formats, these motivations, along with LoRA’s lightweight computational requirements, make it a natural and broadly compatible choice for multi-concept composition.
>
> [1] Gandikota et. al. "Concept Sliders: LoRA Adaptors for Precise Control in Diffusion Models", ECCV 2024
>
> [2] Ruiz et. al. "HyperDreamBooth: HyperNetworks for Fast Personalization of Text-to-Image Models", CVPR 2024
>
> [3] Dravid et. al. "Interpreting the Weight Space of Customized Diffusion Models", NeurIPS 2024
>
> **Q2: It feels like the test data is insufficient, only 15 identity LoRAs. The paper predominantly focuses on facial features, leaving the effectiveness on other categories unclear.**
>
> Thank you for raising this point. We highlight the composition examples with non-human subjects, as it can be seen in Fig. 1(b) (second row, first image for concept `<Tower>`) in the main paper and Fig. 4 in the supplementary material (for concepts `<Royce>, <Cat>, <Lumiva>, <Jacket>`). We focus on face-identity preservation because it involves fine-grained details across the concepts; related studies such as LoRACLR and Orthogonal Adaptation likewise evaluate on a comparable scale (12 identity LoRAs). To further demonstrate generalization beyond human subjects, we trained 18 non-human concept LoRAs from the DreamBooth [4] dataset, formed 10 concept pairs, and generated multi-subject compositions using prompt templates from [5]. We then measured text–image alignment (CLIP-T, HPS), visual quality (Aesthetics), and concept preservation (DINO), comparing LoRAShop with UNO, OmniGen, MS-Diffusion, and MIP-Adapter under the same multi-subject evaluation protocol used in the main paper. The quantitative results are provided in the table below.
>
> | Method        | CLIP-T |  HPS  | Aesthetics |  DINO |
> |---------------|:------:|:-----:|:----------:|:-----:|
> | MS-Diffusion  | 0.483 | 0.291 | 5.609 | *0.446* |
> | MIP-Adapter   | 0.414 | 0.244 | 5.561 | 0.383 |
> | OmniGen       | 0.468 | 0.273 | **5.960** | 0.364 |
> | UNO           | **0.493** | *0.301* | **5.960** | 0.383 |
> | **Ours**      | *0.487* | **0.307** | *5.923* | **0.473** |
>
> As our quantitative results demonstrate, LoRAShop is the only method that consistently ranks first or second across all evaluated multi-subject composition benchmarks. To further illustrate this capability, we will include qualitative examples in the camera-ready version, which could not be provided here due to the formatting constraints of the rebuttal.
>
> [4] Ruiz et. al., "DreamBooth: Fine Tuning Text-to-Image Diffusion Models for Subject-Driven Generation", CVPR 2023
>
> [5] Wu et. al. "Less-to-More Generalization: Unlocking More Controllability by In-Context Generation", ICCV 2025
>
> **Q3: The authors conducted ablation experiments on the mask extraction effect across different layers. Has something like ConceptAttention been attempted? Their method appears to be more robust.**
>
> As an alternative masking strategy, we include quantitative comparisons with ConceptAttention in the table below. Specifically, for the examples used in our Table 4 evaluation, we first extract subject masks using ConceptAttention (based on FLUX-generated outputs), and then perform residual blending with those masks. However, as acknowledged in the original ConceptAttention paper [6], the method has difficulty separating highly similar concepts, such as two different identities, which is precisely the setting of our evaluation. We quote the following limitation from the ConceptAttention paper:
>
> *"The primary limitation of **ConceptAttention** is that it struggles to differentiate between very similar textual concepts. For example, for a photo with a sky with the sun in it, the model does not necessarily know where the boundary of the sun resides, instead capturing a halo around the sun."*
>
> Indeed, in our experiments, we observe that ConceptAttention fails to produce clearly separated masks for different identities, which leads to LoRA cross-talk and diminished identity preservation. The corresponding quantitative results are shown below:
>
> | Masking Strategy      |   ID   | CLIP-T |  HPS  | Aesthetics |
> |-----------------------|:------:|:------:|:-----:|:----------:|
> | w/ ConceptAttention   | 0.362 | 0.250 | **6.189** | 0.261 |
> | **Ours**              | **0.532** | **0.252** | 6.124 | 0.261 |
>
> As shown by the quantitative results, LoRAShop, using the subject priors introduced in our study, successfully separates closely related concepts (such as two distinct identities), leading to superior identity preservation while performing comparably in text–image similarity and visual quality.
>
> [6] Helbling et. al. "ConceptAttention: Diffusion Transformers Learn Highly Interpretable Features", ICML 2025

---

> > ### Comment · Reviewer_ddJF · 2025-08-04
> >
> > Thank you for addressing my concerns. I'm satisfied with the response and will adjust the score accordingly.

---

### Official Review · Reviewer_7KJw · 2025-07-02

**Clarity:** 3
**Significance:** 3
**Originality:** 3
**Rating:** 5
**Confidence:** 3

**Summary:**

This paper introduces LoRAShop, a training-free framework for multi-concept image generation and editing using LoRA adapters in diffusion models. The insight is transformer features in Flux-style diffusion models activate spatially coherent regions early in denoising, enabling the extraction of disentangled subject priors. LoRAShop performs a forward pass to extract coarse masks delimiting concept regions, then selectively blends LoRA adapter features only within these regions during reverse diffusion. This approach eliminates "LoRA cross-talk" without retraining, allowing simultaneous use of multiple adapters. The method supports both generating new multi-concept images and editing existing images by inserting personalized concepts. Extensive experiments demonstrate superior identity preservation over baselines in both generation and editing tasks, with applications ranging from artistic creation to identity transfer beyond simple face swapping.

**Questions:**

1. Generalization Beyond FLUX:
Q: Can LoRAShop’s mask extraction work with convolutional U-Nets (e.g., Stable Diffusion)? If not, what architectural properties are essential?
2. Full-Body Identity Evaluation:
Q: ArcFace only evaluates faces, but Fig. 3 shows body edits. How is full-body identity preserved quantitatively?
3. Computational Overhead:
Q: How much memory does simultaneous multi-adapter blending incur compared to sequential application?
4. Minor issues:
* Typos. 'Expermential Setup We use ... ' -> 'Expermential Setup. We use ... '. L176, P5.

**Ethical Concerns:**

["NO or VERY MINOR ethics concerns only"]

**Final Justification:**

With most concerns addressed, I vote for acceptance.

**Limitations:**

yes

**Quality:**

3

**Strengths And Weaknesses:**

# Strengths
1. This work addresses a critical practical limitation—the inability to combine LoRAs without interference—that has hindered real-world deployment. By solving multi-adapter composition in a training-free manner, it transforms LoRAs into practical "photoshop-like" tools, enabling new creative workflows. The insight about spatial activation patterns in diffusion transformers and the subsequent mask extraction/blending mechanism represents a conceptual leap beyond existing LoRA composition methods. The approach cleverly bypasses the need for joint training or constraints.
2. The method is technically sound and thoroughly validated. Comprehensive experiments cover single/multi-concept generation, real/image editing, and face swapping. Quantitative metrics (ArcFace, CLIP-T, HPS) and user studies demonstrate clear improvements over strong baselines (DreamBooth, IP-Adapter, etc.). Ablation studies justify key design choices (e.g., using the last double-stream block for masks).

# Weaknesses
1. The method relies heavily on Flux’s double-stream/single-stream transformer blocks. While Sec 4.3 ablates block choices, it’s unclear how transferable the approach is to other popular diffusion architectures (e.g., Stable Diffusion’s U-Net). Limited experiments on non-human concepts (e.g., objects/styles) also raise questions about generalizability.
2. Evaluation Limitations: Identity metrics (ArcFace) focus on facial features, but LoRAShop claims full-body editing (Fig. 3). No metric evaluates body-level identity preservation. The editing pipeline uses RF inversion (Sec 3.4), but inversion errors could propagate; this isn’t quantified.

---

> ### Author Rebuttal · Authors · 2025-07-31
>
> We thank reviewer 7KJw for the valuable feedback and finding our method effective and well validated. Please find our responses to the addressed concerns below.
>
> **Q1: Limited experiments on non-human concepts (e.g., objects/styles) also raise questions about generalizability.**
>
> Thank you for drawing attention to the importance of broader experiments for generalizability. We respectfully note that both the main paper and the supplementary material already include non-human examples, for instance, Fig. 1(b) (second row, first example for concept `<Tower>`) in the main paper and Fig. 4 in the supplementary material (for concepts `<Royce>, <Cat>, <Lumiva>, <Jacket>`).
>
> Since identity preservation requires retaining fine-grained details, it poses a particularly challenging generalization task; therefore, we chose this setting for our core multi-subject evaluations. We also point out that related works such as LoRACLR and Orthogonal Adaptation report results on a similar number of identity LoRAs (12 identities). Nonetheless, to demonstrate that our method generalizes beyond human subjects, we have added further quantitative results. We trained 18 non-human concept LoRAs from the DreamBooth dataset, formed 10 concept pairs, and evaluate text–image alignment (CLIP-T, HPS), visual quality (Aesthetics), and concept preservation (DINO). We compared LoRAShop with UNO, OmniGen, MS-Diffusion, and MIP-Adapter, using the same multi-subject evaluation setup as in the main paper. The results are presented in the table below.
>
> | Method        | CLIP-T |  HPS  | Aesthetics |  DINO |
> |---------------|:------:|:-----:|:----------:|:-----:|
> | MS-Diffusion  | 0.483 | 0.291 | 5.609 | *0.446* |
> | MIP-Adapter   | 0.414 | 0.244 | 5.561 | 0.383 |
> | OmniGen       | 0.468 | 0.273 | **5.960** | 0.364 |
> | UNO           | **0.493** | *0.301* | **5.960** | 0.383 |
> | **Ours**      | *0.487* | **0.307** | *5.923* | **0.473** |
>
> As demonstrated by our quantitative evaluations, LoRAShop consistently achieves the best or second-best scores across all criterion, establishing its effectiveness for multi-subject composition. We will include qualitative results for these compositions in the camera-ready version.
>
> **Q2: Generalization Beyond FLUX: Can LoRAShop’s mask extraction work with convolutional U-Nets (e.g., Stable Diffusion)? If not, what architectural properties are essential?**
>
> We rely on the separability property of attention masks in the final double-stream block, as highlighted in our paper. This property of FLUX enables us to successfully blend multiple concepts within the same image, even when their spatial regions overlap (see Fig. 1(a) in the main paper and Fig. 4, third row, in the appendix). However, as long as such masks can be extracted, our method is also applicable to U-Net-based diffusion models, where masked blending of latents has been explored in prior work such as [1].
>
> [1] Avrahami et. al. "Blended Latent Diffusion," SIGGRAPH 2023
>
> **Q3: Identity metrics (ArcFace) focus on facial features, but LoRAShop claims full-body editing (Fig. 3). No metric evaluates body-level identity preservation.**
>
> As pointed out by the reviewer, standard metrics like ArcFace primarily evaluate facial similarity and thus are insufficient for assessing full-body identity preservation quantitatively. To address this limitation, we conducted a user study specifically designed to evaluate full-body identity preservation. In our study, participants were presented with full-body reference images of the subjects represented by the LoRAs (e.g., Billie Eilish in a standing pose), alongside images generated by our method and baseline methods. Participants were then asked to rate how accurately each generated image preserved the full-body identity compared to the reference image, using a scale from 1 to 5 (1 = poor, 5 = excellent). We compare the performance of LoRAShop with OmniGen and InfiniteYou, selected as the strongest baselines according to Table 3 of the main paper. Each participant was shown 40 personalized generations and asked to answer the following question: How well does the "Output" image reflect the full-body physical appearance of the image on the left?
>
> The quantitative results of this user study are summarized in the table below. As shown, our method consistently outperforms baseline approaches in preserving full-body identity. We will include additional qualitative examples and a more detailed analysis in the camera-ready version to further support these findings.
>
> | Method | Pref. Score |
> |--------|:-----------:|
> | OmniGen      | 2.608 |
> | InfiniteYou  | 2.848 |
> | **Ours**     | **3.228** |
>
> **Q4: How much memory does simultaneous multi-adapter blending incur compared to sequential application?**
>
> Implementation-wise, our method applies each LoRA adapter sequentially and does not introduce any parallelism between adapters. Thanks to this sequential design, during inference we iteratively enable and disable LoRA adapters, resulting in GPU memory usage comparable to FLUX without requiring any customization. Specifically, our method uses 36 GB of GPU memory, while FLUX alone requires 32 GB. All experiments are conducted on a single NVIDIA L40 GPU.
>
> **Q5: The editing pipeline uses RF inversion (Sec 3.4), but inversion errors could propagate; this isn’t quantified.**
>
> While we acknowledge that existing inversion methods may introduce errors that propagate across generations, we show in Fig. 3 of the main paper that our method preserves the inverted features without introducing additional divergence from the input. Instead of using RF-Inversion to invert the image latent, we adopt **RF-Edit** [2], as specified in Sec. 3.4, a more robust inversion pipeline that introduces value sharing in the single-stream transformer blocks to enhance content preservation during inversion. Importantly, our approach is compatible with any inversion method, as demonstrated with RF-Edit, since our editing and generation strategy is formulated as a masked residual summation over existing transformer features, which preserves the original model behavior.
>
> To further evaluate the effectiveness of our method on real images, we present quantitative comparisons on the face swapping task in Table 2 of the main paper. Despite potential inversion-related errors, our method matches inpainting-based approaches in content preservation, despite the latter being explicitly restricted from altering unrelated pixels, and surpasses them in target face identity similarity.
>
> [2] Wang et. al. "Taming Rectified Flow for Inversion and Editing", ICML 2025

---

### Official Review · Reviewer_tkqk · 2025-07-02

**Clarity:** 3
**Significance:** 4
**Originality:** 3
**Rating:** 5
**Confidence:** 4

**Summary:**

This paper proposes LoRAShop, a novel training-free framework for multi-concept image generation and editing using pre-trained LoRA. The overall framework first extract disentangled spatial priors for each concept. Then, these priors are then used to guide residual feature blending. Quantitative and qualitative results demonstrate the effectiveness of LoRAShop on various tasks.

**Questions:**

Questions:

1. Could the processing in lines 133–135 potentially lead to additional error?
2. How would the method perform on overlapping concepts (e.g. a couple embracing each other)
3. Could the framework benefit from the learned hyperparameters, such as $\tau$.

**Ethical Concerns:**

["NO or VERY MINOR ethics concerns only"]

**Final Justification:**

My concerns are addressed. I am happy to vote for acceptance.

**Quality:**

3

**Strengths And Weaknesses:**

Strengths:

1. Sufficient Experiments.
2. Generalization. The proposed framework can support various tasks.
3. Novelty. Propose a training-free framework without tuning the LoRA adapters.
4. Practical Impact. The training free nature makes the framework efficient, supports both generated and real-image editing.

Weaknesses:

1. Visual Clarity. Figure 2 can be designed more accurate to illustrate the overall framework.
2. Limitation of Distangled Masks. The extracted mask may inherit biases from the 3x3 Gassian kernel.

---

> ### Author Rebuttal · Authors · 2025-07-31
>
> We thank reviewer tkqk for the valuable feedback and for finding our method *well-evaluated, generalizable, novel,* and *practical*.  Please find our detailed responses to the raised concerns below.
>
> **Q1: Figure 2 can be designed more accurately to illustrate the overall framework.**
>
> We thank the reviewer for the feedback regarding the framework figure. While we are unable to include an updated version in this rebuttal due to formatting constraints, we will revise the figure in the camera-ready version to improve clarity. Additionally, the prior extraction and blending procedures are detailed in Algorithms 1 and 2 in the supplementary material for further reference.
>
> **Q2: The extracted mask may inherit biases from the 3 × 3 Gaussian kernel.**
>
> While we acknowledge that Gaussian kernels may introduce biases, we select them because they emphasize regions with stronger attention values and progressively suppress low-confidence areas over successive iterations. In the camera-ready version, we will illustrate how the mask structure evolves across iterations. Notably, due to this effect of the Gaussian kernels, the subsequent arg-max operation can correctly separate subjects by prioritizing regions with stronger attention responses relative to others.
>
> **Q3: Could the processing in lines 133–135 potentially lead to additional error?**
>
> Thank you for raising this implementation detail. In the processing described in lines 133–135, we apply connected-component analysis to improve efficiency by capping the number of Gaussian filtering steps. Concretely, the loop terminates after at most ten iterations when a single connected component has not yet emerged. We examined the impact of the masking-threshold parameter in the ablation studies shown in Figure 4 of the main paper; as illustrated, choosing a threshold in the 50% – 70% range yields the highest-quality subject priors. We appreciate the opportunity to explain this design choice and will highlight these findings more clearly in the revised manuscript.
>
> **Q4: How would the method perform on overlapping concepts (e.g., a couple embracing each other)?**
>
> Thank you for highlighting this important use case. Our submission already demonstrates LoRAShop’s ability to handle overlapping subjects in both qualitative and quantitative settings. Figure 1(a) of the main paper blends three LoRAs: <Lumiva> (eyeglasses), <DiCaprio> (identity), and <Jacket> (clothing), within the same spatial region, while Appendix Fig. 4 shows additional overlaps such as <Lumiva>, <Messi>, <Jacket> and <Lumiva>, <Armas>, <Jacket>. Appendix Figs. 3 & 4 further depict closely interacting subjects: two identities holding each other around the waist or placing hands on each other’s shoulders, a subject with a LoRA cat on their lap, and another standing directly in front of a LoRA car.
>
> Beyond these visual examples, we conducted a dedicated experiment to quantify performance on overlapping concepts. We generated 50 images with the prompt “a teapot with a face embossment on it, on a dinner table,” which combines a DreamBooth-trained LoRA for the teapot with a pretrained <Pitt> identity LoRA (see Supplementary Table 2 for adapter details). The results, summarized in the accompanying table, show that LoRAShop outperforms MS-Diffusion and OmniGen in preserving both the teapot and face concepts (ID and DINO metrics) while achieving higher text–image alignment (CLIP-T and HPS) and visual coherence (Aesthetics). Full adapter specifications are provided in Supplementary Tables 1–3, and we will elaborate on these capabilities in the revised manuscript.
>
> | Method |  ID  | DINO | CLIP-T |  HPS  | Aesthetics |
> |--------|:----:|:----:|:------:|:-----:|:----------:|
> | MS-Diffusion | 0.308 | **0.645** | 0.528 | 0.298 | 5.534 |
> | OmniGen      | 0.286 | 0.625 | 0.520 | 0.286 | 5.735 |
> | **Ours** | **0.651** | **0.645** | **0.581** | **0.326** | **5.818** |
>
> As shown by the quantitative results, our method outperforms competing approaches in preserving both the teapot and face concepts (evaluated by ID and DINO metrics), while also generating images with higher text–image alignment (evaluated by CLIP-T and HPS scores) and improved visual quality (evaluated by the Aesthetics score).
>
> [1] Ruiz et. al., "DreamBooth: Fine Tuning Text-to-Image Diffusion Models for Subject-Driven Generation", CVPR 2023
>
> **Q5: Could the framework benefit from learned hyper-parameters, such as T?**
>
> Thank you for this insightful suggestion. While our current implementation utilizes fixed hyperparameters (such as the timestep T) for simplicity and ease of inference, exploring learned or adaptive hyperparameters could indeed further enhance the performance and robustness of our framework. We agree that dynamically adjusting hyperparameters like T based on the complexity or characteristics of the input scenario could yield improved results. We will explore this promising direction as future work and discuss its potential benefits clearly in the revised manuscript.

---

> > ### Comment · Reviewer_tkqk · 2025-08-09
> >
> > Thanks for the response. My concerns are addressed. I am happy to maintain my score.

---

### Official Review · Reviewer_o5bb · 2025-07-04

**Clarity:** 3
**Significance:** 2
**Originality:** 2
**Rating:** 5
**Confidence:** 3

**Summary:**

LoRAshop devises a method for applying the combined effect of multiple LORA customziations on the same synthesized image. It operates by narrowing the effect of a specific LORA to a masked region of the generated image.  LORAshop defines a method for automatically deriving a mask from a "subject prior" by running the generation process up to an intermediate timestamp, then ensuring that the mask corresponding to each LORA does not overlap; then subsequently restricting the action of each LORA to a corresponding mask.  The paper conducts evaluations including human evaluations compared to several baseline methods that are capable of combining multiple custom subjects in an image.

**Questions:**

[My original questions are below.  I have revised the rating in light of the author's response.]

* What is the performance of the method (especially compared to non-explicit-masking to baselines) when the multiple customized subjects have overlapping effects? For example, when one subject says is "as shown as a decoration on a vase" and another subject is "the person looks like Brad Pitt"; does the method succeed in drawing Brad Pitt on a vase?
* Some previous mask-oriented methods have previously focused on good performance in the presence of occlusions. Has this paper tested performance on these cases?  How does it do, and how does it compare to the previous occlusion-friendly methods?
* The paper mentions some failure cases. What do these look like?  Why aren't examples shown and discussed in detail?

**Ethical Concerns:**

["NO or VERY MINOR ethics concerns only"]

**Final Justification:**

The authors have answered my main concerns clearly and I have revised my rating to "accept".

**Limitations:**

Although a few words are given about the technical limitations of the paper, the scope of the paper is very limited when compared to baseline methods that, for example, previously combined multiple subjects using open-text instruction descriptions instead of masks (like Omnigen) or that attempted to combine subjects that could overlap and affect the same pixels of the target image (like MS-Diffusion).  This limitation should be explored in more detail, with failure cases comopared to previous mthods explicitly measured and shown. The method seems like it might be superior for a narrow and important use-case, but this narrow scope should be clarified and put in context with previous work that may be superior for other uses.

**Paper Formatting Concerns:**

No issues.

**Quality:**

3

**Strengths And Weaknesses:**

Strengths: the paper's method is simple, understandable. The paper selects a reasonable set of baseline methods to compare to, and benchmarks suggest that it performs well against baselines.

Weaknesses: the paper does not measure performance on difficult multi-subject tasks such as where subjects overlap in the same region of the target image. In particular, by imposing the assumption that different subjects should correspond to non-overlapping masks, the paper implicitly raises the question of whether it suffers from the "bitter lesson" in which its performance win might be too narrowly tailored and will suffer compared to previous methods such as OmniGen and MS-Diffusion that apply learning methods to more flexibly combine mutiple subjects. This potential drawback should be discussed in detail, and measured. The paper does not appear to measure any tasks in which the subjects might overlap. It also does not show any failure cases in detail. Since many of the baseline methods attempt to address these more-difficult cases, the paper should compare its performance on this basis

The paper would be significantly strenthened if it discussed its limited scope in more detail, showing both wins and losses and discussing tradeoffs. Instead, by not investigating difficult cases, and exploring potential disadvantages, it effectively overclaims by implicitly claiming unvarnished superiority over other methods that have previously attempted more difficult forms of the multi-subject customzation task.

---

> ### Author Rebuttal · Authors · 2025-07-30
>
> We thank reviewer o5bb for the valuable feedback and finding our method simple, understandable and well evaluated. Please find our responses to the raised concerns below.
>
> **Q1: The paper does not measure performance on difficult multi-subject tasks such as where subjects overlap in the same region of the target image. (...) The paper does not appear to measure any tasks in which the subjects might overlap.**
>
> We thank the reviewer for this insightful comment. We would like to clarify that our paper and appendix contain multiple examples demonstrating overlapping subjects within the same spatial region (please see details below). We would also like to address any potential misunderstanding: while our method indeed emphasizes non-overlapping masks for clarity in multi-subject compositions, it does not inherently impose a limitation against handling overlapping concepts. Specifically, our method effectively handles overlapping subject regions by performing an attention-driven residual blending during the denoising process, allowing multiple LoRAs to jointly influence the same spatial region without explicit retraining or constraints.
>
> We provide several practical demonstrations of successful overlapping cases in our experiments. For instance, in Fig. 1(a) of the main paper, we combine identity, clothing, and accessory LoRAs` <Lumiva> (eyeglasses), <DiCaprio> (identity), and <Jacket> (clothing)`. Similarly, our Appendix Fig. 4 illustrates further combinations, such as `<Lumiva>, <Messi>, <Jacket>` and `<Lumiva>, <Armas>, <Jacket>`, showcasing robust handling of overlapping LoRAs.
>
> Furthermore, Appendix Figs. 3 and 4 depict closely interacting subjects, featuring scenarios where subjects (each represented by individual LoRAs) hold each other around the waist (Fig 3), place hands on each other's shoulders (Fig 3), hold a LoRA cat on their lap (Fig 4), or stand directly in front of a LoRA car (Fig 4). These examples underscore our method’s capability to effectively handle complex spatial overlaps.
>
> Note that the details about these LoRAs are available in Tables 1, 2, and 3 of our supplementary material. We will elaborate further on our method’s capability to handle overlapping subjects in the revised manuscript.
>
> **Q2: What is the performance of the method (especially compared to non-explicit-masking to baselines) when the multiple customized subjects have overlapping effects? For example, when one subject says is "as shown as a decoration on a vase" and another subject is "the person looks like Brad Pitt"; does the method succeed in drawing Brad Pitt on a vase?**
>
> Thank you for this suggestion. As requested, we performed an experiment on this complex case, involving a teapot LoRA (trained from Dreambooth dataset) and Brad Pitt LoRA.  Our evaluation consists of 50 generated images based on the prompt “a teapot with a face embossment on it, placed on a dinner table,” where both the teapot and the face are represented by corresponding LoRAs.
>
> To thoroughly assess LoRAShop's performance relative to the requested baseline methods (OmniGen and MS-Diffusion), we evaluated the generated images using multiple metrics: ID and DINO scores for concept similarity, HPS and CLIP-T for text-image alignment, and the Aesthetics score for visual coherence. The results are summarized in the table below.
>
> | Method       |   ID  |  DINO | CLIP‑T |  HPS  | Aesthetics |
> |--------------|-------|-------|--------|-------|------------|
> | MS‑Diffusion | 0.308 | **0.645** | 0.528 | 0.298 | 5.534 |
> | OmniGen      | 0.286 | 0.625 | 0.520 | 0.286 | 5.735 |
> | Ours         | **0.651** | **0.645** | **0.581** | **0.326** | **5.818** |
>
> As demonstrated by our experiments, LoRAShop excels at handling these complex scenarios, providing superior identity preservation and accurately representing the target object (teapot).  We also note that unlike approaches like OmniGen and MS-Diffusion, which necessitate full model retraining, LoRAShop works directly at inference time without explicit retraining or additional constraints.
>
> Due to NeurIPS guidelines, we are not permitted to share qualitative results at this review stage; however, visual examples will be provided in the camera-ready version.
>
> **Q3: Some previous mask-oriented methods have previously focused on good performance in the presence of occlusions. Has this paper tested performance on these cases? How does it do, and how does it compare to the previous occlusion-friendly methods?**
>
> We thank the reviewer for raising this important point. As a comparison with occlusion-friendly methods, we provided quantitative evaluations against "Kong et.al., OMG: Occlusion-friendly Personalized Multi-concept Generation in Diffusion Models (2024)". These comparisons, including multi-subject evaluation metrics, are detailed in Table 4 of the main paper, and user study results are presented in Table 1 (main paper). Our method consistently outperforms existing occlusion-friendly methods in both sets of evaluations. Additional qualitative results will be included in the camera-ready version.
>
> Moreover, we provide qualitative comparisons against concept-merging approaches such as Mix-of-Show, Orthogonal Adaptation, Prompt+, and LoRACLR in the supplementary material (Figs. 7, 8, and 9). These qualitative assessments demonstrate that our method produces diverse and occlusion-tolerant compositions without relying on external pose guidance, effectively leveraging the high-quality generative capabilities of the base model, unlike the comparative methods that necessitate external pose inputs to distinctly separate entities.
>
>
> **Q4: The paper mentions some failure cases. What do these look like? Why aren't examples shown and discussed in detail?**
>
>
> Thank you for highlighting this important point. In our initial manuscript, we briefly mentioned technical limitations however we acknowledge the importance of clearly illustrating and discussing failure cases.
>
> Since our method relies on object locations extracted from the attention representations learned by FLUX, the performance of our method is limited by the robustness of the model in multi-subject scenarios. As also mentioned in the official model checkpoints of the Flux.1-dev model, the developers state that “The model may fail to generate output that matches the prompts.” (please see Huggingface’s Flux.1-dev webpage for more details).
>
> Therefore, while our method effectively handles scenarios with up to four subjects (as illustrated in Appendix Fig. 9), accuracy declines when attempting to generate a higher number of subjects (e.g., six or seven). This decrease in accuracy occurs due to FLUX's inherent blending of attentions when numerous subjects are simultaneously generated.
>
> We note that this limitation is a broader ongoing challenge in the recent multi-concept personalization literature. Recent approaches such as LoRACLR, Orthogonal Adaptation, and Mix-of-Show address this by explicitly employing additional conditioning information (e.g., pose or keypoint inputs via ControlNet). In contrast, our approach uniquely handles multi-subject composition without any explicit mask or external conditions.
>
> While we are not allowed to share visual examples due to NeurIPS’s rebuttal guidelines,  we will provide comprehensive qualitative examples and a detailed discussion of these challenging cases in the camera-ready version.
>
>
>
>
> **Q5: The paper would be significantly strengthened if it discussed its limited scope in more detail, showing both wins and losses and discussing tradeoffs. Instead, by not investigating difficult cases, and exploring potential disadvantages, it effectively overclaims by implicitly claiming unvarnished superiority over other methods that have previously attempted more difficult forms of the multi-subject customization task.**
>
> Thanks for the opportunity to clarify the novelty and scope of our work. To the best of our knowledge, our paper introduces the first multi-LoRA 'editing' framework in the literature, distinguishing it from previous methods limited to composing entirely new images. Unlike alternatives such as face-swapping or inpainting, which primarily handle changes restricted to the face region, our approach effectively captures and reflects the full identity associated with LoRAs, including both facial and body-level features. Furthermore, LoRAShop is the first framework capable of performing multi-concept tasks specifically with Flux models.
>
> Moreover, our method addresses complex multi-subject customization tasks in a fully training-free manner. Unlike existing approaches like OmniGen and MS-Diffusion, which necessitate full model retraining, LoRAShop uniquely facilitates flexible multi-concept editing and composition directly at inference time without explicit training or additional constraints.
>
> In addition, unlike recent multi-LoRA composition methods such as LoRACLR, Orthogonal Adaptation and Mix-of-Show, our method does not require additional conditioning information (e.g., pose or keypoint inputs via ControlNet) and handles multi-subject composition without any explicit mask or external conditions.
>
> Our comprehensive evaluations demonstrate LoRAShop’s robust capability to handle challenging scenarios, including overlapping subjects, intricate subject-object interactions, and complex spatial relationships. In our main paper, we provide an extensive number of visual examples showcasing these capabilities (45 examples in total), supplemented by additional examples in our Appendix (75 examples in total).
>
> We recognize the importance of transparently outlining both strengths and limitations. Therefore, in our revised manuscript, we will provide a detailed discussion on trade-offs, explicitly highlighting specific challenging scenarios and comparative strengths to ensure a balanced and thorough assessment of our contributions relative to existing methods.

---

### Note · Authors · 2025-08-14

We thank the reviewers and Area Chair for their insightful feedback, which has strengthened our paper. Reviewers recognized LoRAShop as a **simple, effective, and practical framework** for training-free multi-concept editing and generation. LoRAShop leverages spatial activation patterns in Flux-style transformers to derive concept masks, allowing precise blending of LoRA features to eliminate interference while preserving image context and detail.

Our rebuttal addressed initial reviews with **new experiments and clarifications**:
- **Overlapping Concepts & Occlusions:** We ran new experiments requested by reviewers (e.g., "a teapot with a face embossment on it, placed on a dinner table"), showing superior identity preservation and text alignment over baselines like OmniGen and MS-Diffusion, all without retraining.
- **Generalizability Beyond Faces:** New evaluations on 18 non-human object LoRAs demonstrate strong performance in diverse categories, ranking top-two across all metrics against specialized methods like MIP-Adapter and UNO.
- **Full-Body Identity:** A new user study on full-body identity preservation confirmed LoRAShop outperforms strong baselines like OmniGen and InfiniteYou in maintaining a subject's complete appearance.
- **Superior Masking Strategy:** A requested comparison with ConceptAttention confirmed our method is more robust for separating highly similar concepts, a noted limitation of ConceptAttention, yielding superior identity preservation.

LoRAShop is the first multi-LoRA editing framework for Flux-like rectified flow transformers, operating at inference time without extra constraints like ControlNet or pose guidance; a key distinction from methods like LoRACLR (CVPR'25), Orthogonal Adaptation (CVPR'24), and Mix-of-Show (NeruIPS'23). It extends to real-image editing, with performance matching or exceeding inpainting-based methods in face-swapping benchmarks.

We believe our work provides a practical "photoshop with LoRA adapters" and aligns with the NeurIPS community's goal of building more scalable and controllable generative AI.

---

### Decision · Program_Chairs · 2025-09-17

**Decision:**

Accept (spotlight)

**Comment:**

On this work, the authors introduce LoRAShop, a training-free method that can be used to generate and edit multi-concept images. To do so,  the authors first observe that they can design a disentangled mask extraction technique that can be used to localize the influence of each subject that needs to be personalized, using LoRA technique in diffusion models. This way the authors show how their method can be used to blend multiple concepts in a controlled manner in the diffusion latent. The authors show a bunch of good results both qualitatively and qualitatively to back their claims.

Originally, the reviewers scored the paper 3x Borderline Accept and 1x Borderline Reject. Two of the reviewers asked the authors to further discuss the performance on difficult multi-subject tasks such as where subjects overlap in the same region of the target image. The authors properly address this in the rebuttal (showing experimental results), and had partially addressed this in the original paper. Reviewer 7KJw asks for more extensive results in generalizability, identity metrics and memory performance which was properly addressed. In general, the other weaknesses found were relatively small and properly addressed, except using the method with other diffusion models instead of Flux which the authors could not properly address. Nevertheless, that is a minor issue.

During the authors-reviewers stage, all three positive reviewers increased the score from Borderline Accept to Accept, explicitly mentioning that their concerns have been addressed and thus are happy to raise the score. Reviewer o5bb did not respond to the authors despite the authors and the AC trying multiple times to engage with them. However, after the exchange period, they also increased the score from Borderline Accept to Accept, saying that the authors have addressed their main concerns. Considering that the reviewer was the most senior reviewer, and considering the perfect consensus between the reviewers, there is not any remaining big issue for the authors to address that would block their publication.

To summarize, the AC and the reviewers like the following aspects of the paper:

- The problem the authors address is interesting and well-executed.

- There is sufficient novelty on the paper.

- The paper is well-written.

- The experimental results, be them qualitative and quantitative are very good.

I would urge the authors to use parts of the rebuttal to improve the paper for the camera-ready version. Congratulations to the authors for a very good paper!